# FLiES-SIF ver. 1.0: Three-dimensional radiative transfer model for estimating solar induced fluorescence

Yuma Sakai[1,2], Hideki Kobayashi[1], and Tomomichi Kato[2,3]

[1]Institute of Arctic Climate and Environment Research, Research Institute for Global Change, Japan Agency for Marine-Earth Science and Technology, Yokohama, Japan
[2]Research Faculty of Agriculture, Hokkaido University, Sapporo, Japan
[3]Global Station for Food, Land, and Water Resources, GI-Core, Hokkaido University, Sapporo, Japan

**Correspondence:** Yuma Sakai (ysakai@jamstec.go.jp)

**Abstract.** Global terrestrial ecosystems control the atmospheric $CO_2$ concentration through gross primary production (GPP) and ecosystem respiration processes. Chlorophyll fluorescence is one of the energy release pathways of excess incident lights in the photosynthetic process. Over the last ten years, extensive studies have revealed that canopy scale sun-induced chlorophyll fluorescence (SIF), which potentially provides a direct pathway to link leaf level photosynthesis to global GPP, can be observed from satellites. SIF is used to infer photosynthetic capacity of plant canopy, however, it is not clear how the leaf-level SIF emission contributes to the top of canopy directional SIF. Plant canopy radiative transfer models are the useful tools to understand the mechanism of anisotropic light interactions such as scattering and absorption in plant canopies.. One dimensional (1D) plane parallel layer models (e.g. the Soil Canopy Observation, Photochemistry and Energy fluxes (SCOPE) model) have been widely used and are useful to understand the general mechanisms behind the temporal and seasonal variations in SIF. However, 1D model does not explain the complexity of the actual canopy structures. Three dimensional models (3D) have a potential to delineate the realistic directional canopy SIFs. Forest Light Environmental Simulator for SIF (FLiES-SIF) version 1.0 is a 3D Monte Carlo plant canopy radiative transfer model to understand the biological and physical mechanisms behind the SIF emission from complex forest canopies. The FLiES-SIF model is coupled with leaf-level fluorescence and physiology module so that the users are able to simulate how the changes in environmental and leaf traits as well as canopy structure affect the observed SIF at the top of the canopy. The FLiES-SIF model was designed as three-dimensional model, yet the whole modules are computationally efficient: FLiES-SIF can be easily run by the moderate level of personal computers with lower memory demands and public software. In this model description paper, we focused on the model formulation and simulation schemes, and showed some sensitivity analysis against several major variables such as view angle and leaf area index (LAI). The simulation results show that SIF increases with LAI then saturated at LAI > 2 – 4 depending on the spectral wavelength. The sensitivity analysis also shows that simulated SIF radiation may decrease with LAI at higher LAI domain (LAI > 5). These phenomena are seen in certain sun and view angle conditions. This type of non-linear and non-monotonic SIF behavior to LAI is also related to spatial forest structure patterns. FLiES-SIF version 1.0 can be used to quantify the canopy SIF in various view angles including the contribution of multiple scattering which is the important component in the near infrared domain. The potential use of the model is to standardize the satellite SIF by correcting the bi-directional effect. This step will contribute to the improvement of the GPP estimation accuracy through SIF.

## 1 Introduction

Global terrestrial ecosystems control the atmospheric $CO_2$ concentration through gross primary production (GPP) and ecosystem respiration processes (Canadell et al., 2007; Richardson et al., 2009; Piao et al., 2013). The ecosystem responses to climate
change have not yet been adequately quantified because of insufficient observations and modeling ability (Bunn and Goetz, 2006; Lasslop et al., 2010). Thus, there is great demand in the scientific community for methods of constraining global GPP through existing observation networks (Anav et al., 2015; Teubner et al., 2019). Estimating GPP is essential for various applications, ranging from yield predictions to evaluating and predicting the impact of regional and global environmental changes (Waring et al., 1998; Schimel, 2007).

Chlorophyll fluorescence is an energy release pathway for excess incident light in the photosynthetic process. Over the last ten years, extensive studies have revealed that canopy-scale sun-induced chlorophyll fluorescence (SIF) can be observed from satellites, such as the Greenhouse gases Observation Satellite (GOSAT) (Frankenberg et al., 2011), Orbiting Carbon Observatory-2 (OCO-2) (Li et al., 2018; Norton et al., 2019), Global Ozone Monitoring Experiment-2 (GOME-2) (Joiner et al., 2013), and TROPOspheric Monitoring Instrument (TROPOMI) (Köhler et al., 2018) using Fraunhofer lines in the near-
infrared spectral domain. Satellite-derived SIF potentially provides a direct pathway linking leaf-level photosynthesis to global GPP (Guanter et al., 2010; Frankenberg et al., 2011; Joiner et al., 2013; Porcar-Castell et al., 2014). For example, the observed SIF exhibits a good correlation with net photosynthesis, which is quantified by the monitoring gas exchange method at the leaf level and using the eddy covariance method at the ecosystem scale (Wieneke et al., 2018). SIF can be used to infer the photosynthetic capacity of the plant canopy (Zhang et al., 2018). However, it is not clear how leaf-level SIF emissions contribute
to the top-of-canopy directional SIF, because satellite-observed SIF uses the near-infrared spectral domain, in which multiple scattering on the leaf surface is dominant. Based on the steady-state fluorescence yield theory (Genty et al., 1989), a model for leaf-level SIF and photosynthesis under various environmental conditions has been developed (Van der Tol et al., 2014). The spectral variability of emitted SIF radiance has also been quantified by a radiative transfer model at the leaf level (Agati et al., 1993; Pedrós et al., 2010; Tol et al., 2009), canopy level (Tol et al., 2009; Gastellu-Etchegorry et al., 2017; Yang and van der
Tol, 2018; Liu et al., 2019), and through experiments (Louis et al., 2006; Van Wittenberghe et al., 2015).

    Because of the nonlinear light interactions within plant canopies, the SIF radiance emitted at the top of plant canopies is not simply the sum of the individual leaf contributions (Zeng et al., 2019; Dechant et al., 2020). The top-of-canopy SIF primarily contains fluorescence emissions from sunlit and shaded leaves, and fluorescence signals, which is observed by sensor, enhanced by the multiple scatterings within plant canopies. As most current SIF products from satellites (e.g., GOSAT, GOME-2, OCO-2,
TROPOMI) are derived in the near-infrared spectral domain, where the leaf reflectance and transmittance are high, the multiple-scattering contribution may not be negligible depending on the leaf area (the leaf area index, or LAI). Plant canopy radiative transfer models are useful tools for understanding the mechanism of anisotropic light interactions such as scattering and absorption in plant canopies. One-dimensional (1D) plane parallel layer models (e.g., the Soil Canopy Observation, Photochemistry,

and Energy fluxes (SCOPE) model, Tol et al. (2009)) have been widely used to analyze the physiological, meteorological, and geometrical influences on observed SIF. These plane parallel models provide some insight into the general mechanisms behind the temporal and seasonal variations in SIF. However, the lack of complexity in their actual canopy structures means that 1D models often give inaccurate directional SIF features. Three-dimensional (3D) models (Zarco-Tejada et al., 2013; Gastellu-Etchegorry et al., 2017; Hernández-Clemente et al., 2017), although requiring vast computational resources, have the potential to delineate the realistic directional canopy SIF. The radiative transfer model used in SIF simulations should exhibit several characteristics. First, the contribution of sunlit and shaded leaves to canopy-scale directional SIF emissions should be separately quantified. The intensity of SIF depends on the absorbed photosynthetically active radiation (APAR) on leaf surfaces, and the emissions from sunlit and shaded leaves are quite different (the APAR of sunlit leaves can be 100 times higher than that of shaded leaves). Second, the multiple scattering of fluorescence should be accurately computed, as most satellites use the near-infrared spectral domain. Third, although 3D models are required to evaluate realistic SIF features, the model's input variables should be easily created or accessible from existing databases. This is because, without sufficient input data, it is difficult to extend the model simulations to the various ecosystems around the world. This paper describes a 3D Monte Carlo plant canopy radiative transfer model, the Forest Light Environmental Simulator (FLiES) for simulating canopy-scale directional SIF radiance, and evaluates the performance of the model by analyzing the angular and multiple-scattering effects on SIF.

## 2 Model description

### 2.1 General outline of FLiES-SIF

#### 2.1.1 Overall frameworks

We developed a 3D plant canopy radiative transfer model for simulating the canopy-scale directional SIF radiance (Forest Light Environmental Simulator for SIF, FLiES-SIF version 1.0, Kobayashi and Sakai (2019)). Figure 1(a) shows the overall framework of the FLiES family modules. The aim of FLiES is to consider the impact of landscape heterogeneity on the radiative processes and determine how this links to the canopy energy, water, and carbon exchanges. One of the important aspects of modeling is to make the modules as computationally efficient as possible: most of the FLiES modules, including the newly proposed FLiES-SIF, can be easily run on moderate personal computers with relatively modest memory demands and public software (GNU gfortran, gcc, and R). The model development was initiated using 3D radiative transfer modeling in the shortwave domains (Kobayashi and Iwabuchi, 2008). A 1D version of the atmospheric radiative transfer model, MCARaTS (Iwabuchi, 2006), was incorporated to simulate the atmosphere–forest radiation interaction. A longwave radiative transfer module was then added, together with energy balance and plant physiology modules (Fig. 1(a), Kobayashi et al. (2012); Baldocchi and Harley (1995)). All these modules are related to the radiation emitted in the Stefan–Boltzmann law of the sun, by the earth's surface, and by atmospheric media.

The current FLiES-SIF work adds a radiation interaction module for the induced radiation emitted from leaf pigments, and describes how to combine the forest structure information and leaf physiology models (Van der Tol et al., 2014; Farquhar et al., 1980) with the fluorescence radiative transfer module (Fig. 1(a)). The FLiES-SIF model shares some key aspects of numerical schemes in FLiES: it employs a spatially explicit forest landscape (Sect. 2.1.2) and is based on a Monte Carlo ray-tracing approach (Sects. 2.2– 2.3). Analogous to the modeling in previous FLiES modules, FLiES-SIF employs a Monte Carlo sampling scheme, where photon-tracing sequences represent the integration form of the radiative transfer equation, or the so-called Neumann's series (Antyufeev and Marshak, 1990). In such modeling, the photon path lengths and scattered directions are determined by random numbers and probability distribution functions such as the Lambert–Beer exponential function and the scattering phase function. The scattering and re-absorption of emitted fluorescence light must also be considered to identify the relationship between the fluorescence emitted by the chloroplasts and the top-of-canopy outgoing fluorescence radiance (Porcar-Castell et al., 2014). Several recent studies have worked on the quantification of the impact and modeling of scattering and absorption effects from the leaf scale (e.g., Agati et al. (1993); van der Tol et al. (2019)) to the canopy scale (e.g., Romero et al. (2018)). Multiple scatterings and re-absorption among leaves, trunks, and soil background can be numerically simulated using unbiased and efficient approaches (Kobayashi and Iwabuchi, 2008). The performance and reliability of FLiES for simulating light transmittance through a canopy and bidirectional reflectance factors have been extensively investigated in previous studies (Widlowski et al., 2011, 2013, 2015). As a default setting, FLiES-SIF version 1.0 simulates the bidirectional SIF radiance at the top of the canopy, but the simulation codes can easily be extended to simulate SIF at any height level within the plant canopy and for the spatial mapping of SIF radiance at the top of the canopy.

### 2.1.2 Canopy structure represented by FLiES-SIF

The forest landscapes employed in FLiES-SIF consist of simple geometric objects such as cones, spheroids, and cylinders, as in other 3D models (e.g., DART and FluorFLIGHT) (Fig. 2). The volume domains inside the crown objects can be further split into multiple domains to realize the spatial distribution of the leaf area and woody area densities. This approach is useful in some aspects because (1) it can establish realistic plant canopies in which the majority of leaves are distributed in the outer and upper parts of the crowns (see Fig. 2), and (2) this approach is simple and computationally efficient. The insides of the crown volumes are assumed to be turbid media, where the light attenuation follows the Lambert–Beer exponential law as a function of leaf/woody area densities and leaf inclination angles. The conventional turbid medium approach assumes that the leaves are randomly distributed in space. In the FLiES modules, including FLiES-SIF, a spatially anisotropic arrangement of leaves (the so-called clumping effect) is modeled using re-collision probability theory (Smolander and Stenberg, 2003, 2005; Kobayashi et al., 2010), which is particularly important for the shoot-scale clumping of needle leaf. More details can be found in the FLiES user manual (Kobayashi, 2019). Note that FLiES has a module for the voxel representation of the forest landscape (Wu et al., 2018), but this module is not currently incorporated into FLiES-SIF version 1.0.

The realization of individual tree objects has some degree of freedom in the characterization of leaf and woody densities in crowns. In the sensitivity analysis described in the next chapter, we consider the following forest architectures. The crown objects are separated into two domains, namely the outer leafy-crown and inner woody domains; the outer and inner domains

are filled with 100 % leaves and 100 % wood, respectively. The height and diameter of the inner woody domain is set to be half that of the outer domain (Fig. 2). Stems are represented by solid cylinders. The individual tree dimensions can be defined differently. The landscape size used in this study is $100\,\mathrm{m} \times 100\,\mathrm{m}$. To create the virtual forest canopy for SIF simulations, it is necessary to determine all of the tree positions in the forest. If ground-based tree census data exist, they can be used to create the virtual forest canopy. The virtual forest canopies can also be reconstructed using a statistical approach (Yang et al., 2018). Assuming that the spatial arrangement of the trees follows a Poisson or Neyman distribution, the individual tree positions are determined by these statistical functions and random numbers.

### 2.1.3 Simulation flow

The simulation flow of the spectral SIF calculation is shown in Fig. 1(b). The FLiES-SIF model requires four major inputs, namely geometry data, meteorological data, forest stand data, and optical data for leaves and other elements (e.g., soil background). The geometry data specify the sun and sensor view directions (zenith and azimuth angles). The meteorological data are used in the precomputation of fluorescence yield. The incident total and diffuse photosynthetically active radiation (PAR) data from the top of forest canopies are also used in the canopy radiative transfer module. If no PAR observations are available, these data can be calculated by the FLiES atmosphere module (1D MCARaTS, see Fig. 1(a)).

Before running the Monte Carlo ray-tracing processes in the forest canopy, the forest structures (leaf area density and leaf voxel look-up table) and leaf-level physiology (fluorescence module) are computed. In FLiES-SIF, landscape-scale LAI is an input variable. The model requires the leaf area density of individual canopy volumes. FLiES-SIF computes the leaf area density from a given forest landscape and LAI data. Fluorescence ray-tracing also requires information about the leaf-level sun-induced fluorescence yield and its spectral composition. This information is computed by the leaf photosynthesis and fluorescence module. These modules are described in the following subsections (Sects. 2.1.4 and 2.1.5).

Once the forest structure and leaf fluorescence yield have been computed, Monte Carlo ray tracing is performed in the broad PAR domain to determine the PAR absorbed in the forest landscape ($\mathrm{APAR_c}$ in Sect. 2.2). The SIF radiance is then simulated on an individual wavelength basis. Details of the radiative transfer formulation and ray-tracing algorithm are summarized in the following sections (Sects. 2.2 – 2.5).

### 2.1.4 Creation of the leafy-canopy voxel look-up table

In FLiES-SIF, 3D forest landscapes are reconstructed using the geometric tree objects described in Sect. 2.1.2 and Fig. 3-upper). The photon tracing starts from an arbitrary position $\boldsymbol{v}_0 = (x,\ y,\ z)$ within the leafy-canopy volume. This position is determined by three random numbers corresponding to $x$, $y$, and $z$. When the canopy landscape is sparse, the majority of randomly determined positions $\boldsymbol{v}_0$ will be outside of the leafy-canopy space, which means a large number of trial runs will be required to determine an appropriate position $\boldsymbol{v}_0$. To reduce the computation time, regularly placed leafy-canopy voxel tables are extracted to determine where to start the SIF emission and subsequent photon tracing (Fig. 3). In FLiES-SIF version 1.0, the leafy canopy voxel information is saved in a look-up table (Fig. 3). The voxel information in the table contains lower and upper corner positions ($x, y, z$ and $x + dx, y + dy, z + dz$), the leaf area density (LAD) of the voxel, and the sunlit leaf area density

(LAD$_{\text{sun}}$). The size of each voxel is currently $1\text{m}^3$. Note that the extracted leafy voxels are not always completely filled with canopy geometry—canopy edge voxels only partially contain the leafy-canopy geometry. In addition, tree canopy geometries contain branch domains. Thus, even if the voxel is completely inside the canopy geometry, there may be some domains that do not contain leaves.

### 2.1.5 Computation of leaf-level fluorescence yield

In FLiES-SIF version 1.0, the leaf-level sun-induced fluorescence yield (hereafter SIF yield $\phi_f$) is pre-computed and stored in a look-up table prior to ray tracing. The SIF yield $\phi_f$ is computed by the model of Van der Tol et al. (2014) and Farquhar's photosynthesis model (Farquhar et al., 1980) under various environmental conditions, including absorbed PAR (APAR). The actual photosynthesis can not be determined without stomata models (e.g., Collatz et al. (1991)). The leaf temperature is also dependent on photosynthesis and stomata regulations. These interrelations are solved by iterating the energy balance, stomata, and photosynthesis equations. The CANOAK-FLiES module (Fig. 1(a), Kobayashi et al. (2012)) can handle such leaf-level coupled physiology phenomena, but this would require more input variables and increase the computational load. Thus, in the current FLiES-SIF, we adapted the following assumptions to obtain reasonable photosynthesis simulation results. First, the leaf temperature was assumed to be the same as the surface air temperature. This is usually acceptable, except in very dry conditions when the stomata are almost closed in daytime. The other assumption concerns the stomata modeling. The FLiES-SIF module does not explicitly use the stomata model. Rather, the consequences of the stomata activity, i.e., the down-regulation of the intercellular partial $CO_2$ pressure (ipCO$_2$), were modeled as a function of the vapor pressure deficit (VPD). We used the experimental relationships measured by Dang et al. (1997), who investigated the relationships between ipCO$_2$ and VPD in three tree species (pine, spruce, and aspen, see Fig. 10 of Dang et al. (1997)). If we simulate SIF over such species, the regression lines given by Dang et al. (1997) can be used. For other species, we created a simplified function based on the relationship derived by Dang et al. (1997):

$$\text{ipCO}_2/\text{apCO}_2 = 0.8 \qquad\qquad (0 < \text{VPD}\,(\text{kPa}) \leq 1.0) \tag{1a}$$

$$\text{ipCO}_2/\text{apCO}_2 = -0.2\text{VPD} + 1.0 \qquad\qquad (1.0 < \text{VPD}\,(\text{kPa}) \leq 3.5) \tag{1b}$$

$$\text{ipCO}_2/\text{apCO}_2 = 0.3 \qquad\qquad (3.5 < \text{VPD}\,(\text{kPa})) \tag{1c}$$

where apCO$_2$ denotes the ambient partial $CO_2$ pressure.

To simulate the spectral SIF, the spectral composition of SIF must be known. Our approach is similar to that used in the SCOPE model (Tol et al., 2009). We derived the spectral composition from the FluorMODleaf model (Zarco-Tejada et al., 2006; Pedrós et al., 2010). The calculated leaf-level spectral SIF radiance variations given by FluorMODleaf were normalized to determine the fraction of SIF at wavelength $\lambda$, $f_s$ $\left(\text{mWm}^{-2}\text{sr}^{-1}\right)$, with respect to the broadband SIF $\left(\text{Wm}^{-2}\right)$. The standard-setting and variables are described in Sect. 3.1. That is, we only used the fraction of spectral composition from the FluorMODleaf model. The radiance was then determined from APAR and $\phi_f$, which varies with environmental conditions and leaf traits such as the maximum carboxylation capacity, $V_{\text{cmax}}$, used in the photosynthesis model.

## 2.2 Bidirectional SIF radiance

The bidirectional SIF radiance at wavelength $\lambda$ at the top of canopy, $I(\lambda, \mathbf{\Omega}_v)$, can be decomposed into four different light transfer pathways:

$$I(\lambda, \mathbf{\Omega}_v) = I_{\text{dir\_sun}}(\lambda, \mathbf{\Omega}_v) + I_{\text{dir\_shade}}(\lambda, \mathbf{\Omega}_v) + I_{\text{ms\_sun}}(\lambda, \mathbf{\Omega}_v) + I_{\text{ms\_shade}}(\lambda, \mathbf{\Omega}_v) \tag{2}$$

where the subscripts "dir" and "ms" indicate the direct emission of SIF and SIF after multiple scatterings, respectively. The direction vector $\mathbf{\Omega}_v = (\theta_v, \phi_v)$ contains the observation zenith and azimuth angles. The radiance elements $I_{\text{dir\_sun}}$, $I_{\text{dir\_shade}}$, $I_{\text{ms\_sun}}$, and $I_{\text{ms\_shade}}$ on the right-hand side of Eq. (2) indicate direct SIF radiance from sunlit leaves, direct SIF radiance from shaded leaves, sunlit SIF radiance after multiple scatterings, and shaded SIF radiance after multiple scatterings, respectively. Here, the direct emission of SIF indicates SIF that is emitted from leaves and directly escapes from the canopy space without hitting other leaves and trunks. On the contrary, "multiple scattering SIF" indicates SIF that is emitted from leaves, hits other leaves, trunks, or soil background, and then escapes from the canopy space in the view direction. Note that most of the optical and radiance quantities described below are spectral variables. For simplicity of the mathematical expressions, if not explicitly mentioned, the wavelength $\lambda$ is omitted from subsequent equations.

The intensity of SIF is related to the absorbed photosynthetically active radiation (APAR) taken in by the forest canopy. If the forest is sparse or the leaf area density in the tree crowns is low, a large portion of incident PAR is transmitted through plant canopies. The transmitted PAR does not contribute to the SIF emissions on the leaf surface. Thus, if photon tracing is performed under sparsely vegetated canopies, the simulation includes large amounts of photons that are not used to compute SIF. To make the numerical simulation more efficient, we propose a variance reduction technique. FLiES-SIF forces all incident PAR to be absorbed by sunlit or shaded leaves and initiates the photon tracing for SIF emitted from leaves. This procedure artificially enhances or diminishes APAR, biasing the simulated SIF depending on the ratio of actual APAR to the "apparent APAR" ($\text{APAR}_{\text{app}}$) used in the simulation. Thus, the simulated SIF under the $\text{APAR}_{\text{app}}$ is adjusted to the actual APAR ($\text{APAR}_c$) conditions:

$$I(\mathbf{\Omega}_v) = \frac{\text{APAR}_c}{\text{APAR}_{\text{app}}} I'(\mathbf{\Omega}_v) \tag{3}$$

where $I(\mathbf{\Omega}_v)$ and $I'(\mathbf{\Omega}_v)$ denote the SIF radiance with $\text{APAR}_c$ and $\text{APAR}_{\text{app}}$, respectively. $\text{APAR}_{\text{app}}$ is simulated with the SIF simultaneously. The $\text{APAR}_c$ is independently calculated for a given canopy landscape before the SIF simulation (Fig. 1(b) and Sect. 2.1.3). In subsequent sections, we describe the radiance components derived with $\text{APAR}_{\text{app}}$ ($I'_{\text{dir\_sun}}$, $I'_{\text{dir\_shade}}$, $I'_{\text{ms\_sun}}$, and $I'_{\text{ms\_shade}}$).

## 2.3 Calculation of direct SIF radiance

The direct SIF radiance from sunlit and shaded leaves is calculated by summing all direct SIF radiation contribution factors of the $i$-th photon ($\psi_{\text{dir},i}$):

$$I'_{\text{dir\_sun}} = \frac{1}{N} \sum_{i=1}^{N} \begin{cases} \psi_{\text{dir},i} & \boldsymbol{v}_0 \in V_{\text{sun}} \\ 0 & \boldsymbol{v}_0 \in V_{\text{shade}} \end{cases} \tag{4a}$$

$$I'_{\text{dir\_shade}} = \frac{1}{N} \sum_{i=1}^{N} \begin{cases} 0 & \boldsymbol{v}_0 \in V_{\text{sun}} \\ \psi_{\text{dir},i} & \boldsymbol{v}_0 \in V_{\text{shade}} \end{cases} \tag{4b}$$

where $V_{\text{sun}}$, $V_{\text{shade}}$, $\boldsymbol{v}_0$, and $N$ indicate the classes of sunlit and shaded leaves, the position of the photon $(x, y, z)$, and the total number of photons, respectively.

The direct SIF radiation contribution factor of the $i$-th photon $\psi_{\text{dir},\,i}$ can be decomposed into three components: leaf-level SIF emission weight $w_0$, directional emission transfer function (the so-called phase function $P_f$), and attenuation function:

$$\psi_{\text{dir}} = \frac{w_0 P_f (\boldsymbol{\Omega}_{\text{L}}, \boldsymbol{\Omega}_{\text{v}}) \exp(-\tau_{\text{v}})}{2\pi |\cos\theta_{\text{v}}|} \tag{5}$$

Here, $\tau_{\text{v}}$ is the optical thickness of the plant canopy in the view direction $\boldsymbol{\Omega}_{\text{v}}$. The factor $2\pi$ is a normalization factor for the phase function $P_f$. These three components in $\psi_{\text{dir}}$, namely $w_0$, $P_f$, and $\exp(-\tau_{\text{v}})$, indicate the SIF emitted in all directions from both adaxial and abaxial sides of a single leaf, the fraction of SIF emitted in the view direction, and the fraction of SIF attenuation to the top of the canopy in the view direction, respectively.

### 2.3.1 Attenuation function

The attenuation of SIF in the view direction $\boldsymbol{\Omega}_{\sigma}$ is calculated by the attenuation function $\exp(-\tau_{\text{v}})$. When the hotspot effect is not considered, the attenuation function is expressed using the plant canopy gap fraction theory:

$$\exp(-\tau_{\sigma}) = \exp\left(-\sum_i u_i \gamma_i G_{\sigma,i} s_i\right) \tag{6}$$

where $u_i$, $s_i$, $G_{\sigma,i}$ and $\gamma_i$ are the leaf area density, path length, mean leaf projection area, and clumping index of the $i$-th tree. They are aggregated over the trees located in the light path between the emission point to the top of canopy in the view direction, respectively. The path length, $s$, is a sum of canopy paths that penetrates through crown objects.

The mean leaf projection area $G$ is a function of the leaf inclination angle distribution function $g_{\text{L}}$ and an arbitrary direction $\boldsymbol{\Omega}_{\sigma}$ (such as the sun direction $\boldsymbol{\Omega}_{\text{s}}$ or view direction $\boldsymbol{\Omega}_{\text{v}}$):

$$G_{\sigma} := G(\boldsymbol{\Omega}_{\sigma}) = \frac{1}{2\pi} \int_{2\pi} g_{\text{L}}(\boldsymbol{\Omega}_{\text{L}}) |\boldsymbol{\Omega}_{\text{L}} \cdot \boldsymbol{\Omega}_{\sigma}| \, d\boldsymbol{\Omega}_{\text{L}} \tag{7}$$

Generally, the clumping index contains various nonrandom scales of spatial leaf distributions, from the shoot to the landscape scale. Because FLiES-SIF version 1.0 employs explicit tree crown landscapes, clumping larger than the crown scale need not

be considered. However, the crown volumes are expressed as turbid media: if the leaves are not randomly distributed in the crown object, e.g., shoot-scale clumping (Cescatti and Zorer, 2003; Chen et al., 1997), attenuation must be corrected according to the shoot-scale clumping index. In FLiES-SIF version 1.0, the shoot-scale clumping index is estimated by the spherically averaged shoot silhouette area (Cescatti and Zorer, 2003). Details on how shoot-scale clumping is incorporated can be found in a previous report (Kobayashi et al., 2010). The hotspot effect refers to the strong illumination near the solar direction ($\mathbf{\Omega}_v \approx -\mathbf{\Omega}_s$). When the hotspot effect is nonnegligible, the modified optical thickness $\tau'$ is expressed as:

$$\tau' = \tau H \tag{8}$$

where $H$ is a hotspot function expressed by the Hapke model (Hapke, 2012), which is used in the framework of the FLiES model (Kobayashi and Iwabuchi, 2008):

$$H(\mathbf{\Omega}_L, \mathbf{\Omega}_j) \simeq 1 - \frac{1}{\left(1 + \frac{1}{h(\mathbf{\Omega}_L, \mathbf{\Omega}_j)} \tan(\frac{\alpha_j}{2})\right)} \tag{9}$$

$$h(\mathbf{\Omega}_L, \mathbf{\Omega}_j) \simeq \frac{ul}{2}\left(\frac{G(\mathbf{\Omega}_L) + G(\mathbf{\Omega}_j)}{2}\right) \tag{10}$$

where $\mathbf{\Omega}_j$, $l$, and $\alpha_j$ indicate the incident direction after the $j$-th scattering, the radius of the disk-shaped flat leaves, and the scattering angle ($\alpha_j = \cos^{-1}|\mathbf{\Omega}_v \cdot \mathbf{\Omega}_j|$), respectively.

### 2.3.2 Leaf-level SIF emission weight

The leaf-level SIF emission weight $w_0$ can be calculated from the SIF yield $\phi_f$ and APAR on the leaf surface ($\mathrm{APAR}_L$):

$$w_0 = f_s \phi_f \mathrm{APAR}_L \tag{11}$$

where $f_s$ is the fraction of SIF at wavelength $\lambda$ ($\mathrm{mW\,m^{-2}\,sr^{-1}}$) with respect to the broadband SIF ($\mathrm{W\,m^{-2}}$). Thus, $f_s$ is a function of wavelength. The SIF yield $\phi_f$ is a function of $\mathrm{APAR}_L$ and various environmental and leaf trait variables such as ambient air temperature, humidity, $CO_2$ concentration, and carboxylation capacity (Van der Tol et al., 2014). In FLiES-SIF version 1.0, $\phi_f$ is read from a look-up table across a wide range of $\mathrm{APAR}_L$, which should be pre-computed by the leaf-level SIF yield models.

The exact computation of $\mathrm{APAR}_L$ under the angular dependency of PAR can be performed by backward ray tracing at the given position of a leaf, but this approach is time-consuming. For more efficient simulations, the values of $\mathrm{APAR}_L$ for sunlit and shaded leaves are approximated as the product of the incident-diffuse PAR and the attenuation function $\exp(-\tau s)$ integrated over the upper hemisphere:

$$\mathrm{APAR}_L = \begin{cases} (1 - \omega_{PAR})\left\{\mathrm{PAR}_{dir}|\mathbf{\Omega}_s \cdot \mathbf{\Omega}_L| + \mathrm{PAR}_{dif}\frac{1}{\pi}\int_0^{2\pi}\int_0^{\frac{\pi}{2}}\exp(-\tau s(\theta, \phi))\sin\theta d\theta d\phi\right\} & \text{if } \boldsymbol{v}_0 \in V_{sun} \\ (1 - \omega_{PAR})\mathrm{PAR}_{dif}\frac{1}{\pi}\int_0^{2\pi}\int_0^{\frac{\pi}{2}}\exp(-\tau s(\theta, \phi))\sin\theta d\theta d\phi & \text{if } \boldsymbol{v}_0 \in V_{shade} \end{cases} \tag{12}$$

where $\mathrm{PAR}_{dir}$ and $\mathrm{PAR}_{dif}$ denote the incident direct and diffuse PAR, respectively, $\omega_{PAR}$ is the average single-scattering albedo in the PAR spectral domain ($400 - 700\,\mathrm{nm}$), and $\omega_{PAR}$ is the sum of the leaf reflectance $r_{PAR}$ and transmittance $t_{PAR}$ in the PAR

domain ($\omega_{\mathrm{PAR}} = r_{\mathrm{PAR}} + t_{\mathrm{PAR}}$). This equation assumes that diffuse PAR is isotropic over the sky and neglects direct PAR scattered within the plant canopy and soil background. Thus, $\mathrm{APAR_L}$ may be underestimated when the background reflectance is high, such as in the case of snow cover. To further reduce the computation time, the hemispherical integration of the attenuation function is approximated by an average of the limited-angle samplings. Details of the computation method are given in Sect. 2.5.

### 2.3.3 Phase function for SIF emissions

The phase function for SIF emissions $P_f$ gives the fraction of SIF emitted in the view direction $\mathbf{\Omega}_{\mathrm{v}}$. Similar to the scattering phase function for the reflection of solar illumination, $P_f$ can be determined by the following equations:

$$
P_f\left(\mathbf{\Omega}_{\mathrm{L}}, \mathbf{\Omega}_{\mathrm{v}}\right) = \begin{cases} f_{\mathrm{ada}}\left|\mathbf{\Omega}_{\mathrm{L}} \cdot \mathbf{\Omega}_{\mathrm{v}}\right| & \text{if } \left(\mathbf{\Omega}_{\mathrm{L}} \cdot \mathbf{\Omega}_{\mathrm{s}}\right)\left(\mathbf{\Omega}_{\mathrm{L}} \cdot \mathbf{\Omega}_{\mathrm{v}}\right) > 0 \\ f_{\mathrm{aba}}\left|\mathbf{\Omega}_{\mathrm{L}} \cdot \mathbf{\Omega}_{\mathrm{v}}\right| & \text{if } \left(\mathbf{\Omega}_{\mathrm{L}} \cdot \mathbf{\Omega}_{\mathrm{s}}\right)\left(\mathbf{\Omega}_{\mathrm{L}} \cdot \mathbf{\Omega}_{\mathrm{v}}\right) \leq 0 \end{cases} \tag{13}
$$

where $f_{\mathrm{ada}}$ and $f_{\mathrm{aba}}$ are the fraction of SIF emissions from the adaxial and abaxial sides of a leaf; $f_{\mathrm{ada}} + f_{\mathrm{aba}} = 1$. Note that, in 280 our definition, we have assumed that illumination by solar beams is always on the adaxial side of a leaf.

### 2.4 Multiple scattering

SIF emissions from the leaf surface occur in all directions (upward and downward in the plant canopy), although they are not always isotropic, as shown in Sect. 2.3.3. A certain portion of SIF does not directly go toward the sky. This portion hits other leaves, trunks, or soil background. The SIF energy from those impacts is scattered, goes in another direction, and then impacts 285 something else. We define this process as the multiple scatterings of SIF. After multiple scatterings, some of the SIF energy will return to the view direction, which enhances the observed SIF radiance depending on the magnitude of the multiple-scattering contribution. The multiple-scattering process of SIF is the same as the scattering process of solar radiation, and the multiple-scattering component can be formulated in exactly the same way as the bidirectional reflectance factor described in Kobayashi and Iwabuchi (2008). The scattered SIF radiance emitted by sunlit and shaded leaves is defined as $I_{\mathrm{ms\_sun}}$ and 290 $I_{\mathrm{ms\_shade}}$, respectively, and these radiance contributions can be calculated by summing all of the scattering contributions:

$$
I_{\mathrm{ms\_sun}} = \frac{1}{N} \sum_{i=1}^{N} \sum_{j=1}^{M} \begin{cases} \psi_{i,j} & \boldsymbol{v}_j \in V_{\mathrm{sun}} \\ 0 & \boldsymbol{v}_j \in V_{\mathrm{shade}} \end{cases}
$$

$$
I_{ms\_shade} = \frac{1}{N} \sum_{i=1}^{N} \sum_{j=1}^{M} \begin{cases} 0 & \boldsymbol{v}_j \in V_{\mathrm{sun}} \\ \psi_{i,j} & \boldsymbol{v}_j \in V_{\mathrm{shade}} \end{cases} \tag{14}
$$

Here, $\psi_{i,j}$ is calculated as follows:

$$
\psi_{i,j} = \frac{w_{i,j} P\left(\mathbf{\Omega}_j, \mathbf{\Omega}_{\mathrm{v}}\right) \exp\left(-\tau_{\mathrm{v}}\right)}{4\pi \left|\cos\theta_{\mathrm{v}}\right|} \tag{15}
$$

where $w_{i,j}$ is the weight of the $i$-th photon after the $j$-th scattering obtained by using the single-scattering albedo in the SIF spectral domain $\omega_{\text{SIF}} = r_{\text{SIF}} + t_{\text{SIF}}$ ($w_{i,j} = w_{i,j-1}\omega_{\text{SIF}}$). Equation (15) is exactly the same as the multiple scatterins in the shortwave radiative transfer (Kobayashi and Iwabuchi, 2008). The form of the phase function $P(\mathbf{\Omega}_j, \mathbf{\Omega}_v)$ is also described by Eq. (7) in Kobayashi and Iwabuchi (2008). The attenuation function is the same as described in Sect. 2.3.1.

## 2.5  Photon tracing algorithm

The numerical scheme of the photon tracing is shown in Fig. 4. The procedures framed by the dotted grey rectangle indicate the photon tracing scheme for direct SIF emissions. The area outside the dotted grey rectangle corresponds to scattered photon tracing. The algorithm for scattered photon tracing is exactly the same as the photon tracing method for solar radiation. Here, we focus on the SIF emission scheme in the grey rectangle. Details of the scattered components are summarized in Kobayashi and Iwabuchi (2008).

### A. Set a new photon in the leafy-canopy

The position $\mathbf{v}_0 = (x, y, z)$ from which SIF emission occurs within a leafy-canopy domain is determined by random numbers. The position $\mathbf{v}_0$ is determined as follows. First, an arbitrary voxel is chosen at random from the voxel table (Fig. 3). The exact position $(x, y, z)$ within a selected voxel is then determined by three random numbers ($Rx$, $Ry$ and $Rz$; $R \in [0,1]$):

$$x = x_l + R_x dx \tag{16a}$$

$$y = y_l + R_y dy \tag{16b}$$

$$z = z_l + R_z dz \tag{16c}$$

where $\mathbf{v}_l = (x_l, yl, zl)$ denotes the position of the lower corner of the selected voxel. If the selected voxel is an edge voxel or contains branch domains, the randomly determined position $\mathbf{v}_0$ may be outside the leafy canopy. Therefore, the position $\mathbf{v}_0$ is checked to determine whether it is in the leafy domain. If the position is outside the leafy domain, the program generates a new random number and selects another voxel. This procedure continues until the leafy canopy position $\mathbf{v}_0$ is obtained.

### B. Determination of the leaf properties for SIF emission

After position $\mathbf{v}_0$ has been determined, the leaf properties at the selected position are determined. Two leaf properties are required to continue the computation of the SIF emission: the leaf illumination status (sunlit or shaded) and the leaf surface normal vector $\mathbf{\Omega}_{\text{L}} = (\theta_{\text{L}}, \phi_{\text{L}})$. The sunlit leaf area fraction $P_{\text{sun}}$ at $\mathbf{v}_0$ is computed using the interception of direct sunlight:

$$P_{\text{sun}} = \frac{1}{G_{\text{s}}} \lim_{\Delta L \to 0} \frac{\exp(-G_{\text{s}}\gamma L_p) - \exp(-G_{\text{s}}\gamma(L_p + \Delta L_p))}{\Delta L}$$

$$= \gamma \exp(-G_{\text{s}}\gamma L_p) \tag{17}$$

where $L_p$ is the cumulative LAI at $\boldsymbol{v}_0$ along the path of the sunlight and $G_{\mathrm{S}}$ is a mean leaf projection area defined in Eq. 7. The leaf illumination status (sunlit or shaded) is then determined by a random number $R$:

$$
\begin{cases}
R \le P_{\mathrm{sun}} & \rightarrow \text{Sunlit leaf} \\
R > P_{\mathrm{sun}} & \rightarrow \text{Shade leaf}
\end{cases}
\tag{18}
$$

The leaf surface normal vector $\boldsymbol{\Omega}_{\mathrm{L}}$ is also required because the leaf-level SIF emission is related to APAR at the leaf surface (APAR$_{\mathrm{L}}$). APAR$_{\mathrm{L}}$ is computed from the cosine of the sunlight and leaf normal angles. Assuming the leaves are randomly

distributed, the azimuthal angle of the leaf surface normal $\phi_{\mathrm{L}}$ can be determined by:

$$
\phi_{\mathrm{L}} = 2\pi R
\tag{19}
$$

For a given leaf angle distribution function $g_{\mathrm{L}} := g(\theta_{\mathrm{L}})$, the zenith angle of the leaf surface normal $\theta_{\mathrm{L}}$ can be determined by the rejection method. In the first step, $\theta_{\mathrm{L}}$ is calculated using a random number:

$$
\theta_{\mathrm{L}} = \frac{\pi}{2} R
\tag{20}
$$

Then, $\theta_{\mathrm{L}}$ is further evaluated using $g_{\mathrm{L}}$:

$$
\begin{cases}
R \le g_{\mathrm{L}} \sin \theta_{\mathrm{L}} & \rightarrow \text{select} \\
R > g_{\mathrm{L}} \sin \theta_{\mathrm{L}} & \rightarrow \text{reject}
\end{cases}
\tag{21}
$$

If $\theta_{\mathrm{L}}$ is rejected by the abovementioned criteria in Eq. (21), the program returns to Eq. (20) and calculates another $\theta_{\mathrm{L}}$. In Eq. (21), the evaluation function is a form of leaf angle distribution function multiplied by a sine value. This sine comes from the Jacobian of the polar coordinate and is necessary because $g_{\mathrm{L}}$ is defined in polar coordinates.

### C. Compute the leaf-level SIF emission and the direct SIF radiance in the view direction

Once the position $\boldsymbol{v}_0$ and leaf properties have been determined, the leaf-level SIF emission $w_0$ and the direct SIF radiance ($I_{\mathrm{dir\_sun}}$ and $I_{\mathrm{dir\_shade}}$) can be computed using the equations derived in Sects. 2.2 and 2.3. The calculation of w0 includes the spherical integration of the attenuation function (Eqs. (11) and (12)), which is time-consuming. Thus, FLiES-SIF version 1.0 approximates this spherical integration by taking the average of five directions $(\theta, \phi) = (0°, 0°), (60°, 0°), (60°, 90°), (60°,$

$180°)$, and $(60°, 270°)$. We tested the performance of this 5-angle assumption by comparing with 10-degree interval samplings (9 zenith and 36 azimuth angles = 324 angle sampilngs). When the attenuation functions were computed by these two angle samplings at 104 randomly selected positions in the forest landscapes used in the sensitivity analysis in section 3, the mean absolute error of this approximation was 14.6 % (N = 10000). Finally, $I_{\mathrm{dir\_sun}}$ and $I_{\mathrm{dir\_shade}}$ are calculated by the local estimation method using Eqs. (4) and (5) (Antyufeev and Marshak, 1990; Marchuk et al., 1980).

### D. Determination of the new emission direction

Direct SIF radiance in the view direction $\boldsymbol{\Omega}_{\mathrm{v}}$ is determined by procedure C. The multiple scattering contribution is further evaluated by photon tracing. To start the photon tracing, the emission direction $\boldsymbol{\Omega}(\theta, \phi)$ is calculated using two random

numbers and the leaf surface normal vector $\mathbf{\Omega}_{\mathrm{L}} = (\theta_{\mathrm{L}}, \phi_{\mathrm{L}})$. Assuming that the SIF emission is bi-Lambertian on the leaf surface, the zenith and azimuthal angles relative to the leaf normal ($\alpha$, $\beta$) are determined by:

$$\alpha = \cos^{-1}\sqrt{R} \qquad\qquad (22)$$

$$\beta = 2\pi R \qquad\qquad (23)$$

The scattering direction $\mathbf{\Omega}(\theta, \phi)$ in the Cartesian coordinate system is then calculated by a coordinate transformation from ($\alpha$, $\beta$) to ($\theta$, $\phi$).

## 3    Sensitivity analysis

We created a one-hectare virtual forest as a default conditions (Fig. 5 (a)). Test simulations of the SIF emissions were performed to evaluate the FLiES-SIF model performance and understand the sensitivity of SIF against various input variables including forest structures. We evaluated the model by four step exercises. First, we conducted the inter-comparisons with the existing 3D model (Discrete Anisotropic Radiative Transfer, DART) to quantify the inter-model differences (Sect, 3.2). Secondly, we performed the sensitivity analysis against geometric conditions (solar zenith angle, SZA; view zenith angle, VZA), sunlit leaf fraction, and LAI, and to identify the factors (hotspots, light attenuation, phase function, weight of photons) that contribute to SIF radiance under the given forest structure (Sect. 3.3). Thirdly, we ran the FLiES-SIF model with different forest landscapes to show how the 3D forest structures such as crown shape, tree size and crown covers influences the simulated SIF (Sect. 3.4). Fourthly, we performed 1D and 3D comparisons in an actual diurnal variations in total and diffuse PAR observed in Yokohama, Japan (35° 22'N, 139° 37'E) in the summer of 2014 (Delta-T sunshine sensor, Delta-T Co Ltd.) (Sect. 3.5). In this exercise, we evaluated the potential errors (overestimation) in the 1D homogeneous layer (turbid medium) approach. We compared in seven different scenarios that include different leaf angle distributions (spherical, erectrophile, and planophie) and within-crown clumping ($\gamma = 0.6$) in the 3D landscapes (Table 1). Lastly, we evaluated the sensitivity of the leaf level fluorescence yield (Sect. 3.6). In this exercise, we computed the leaf level fluorescence with the FLiES-SIF physiology module (Fig. 1(b)).

### 3.1    Input data and simulation condition

The individual tree positions and sizes were determined at random. The spheroid shape was employed for the individual crowns. The tree density used in the sensitivity analysis was 359 trees ha$^{-1}$. The canopy layer height was set to 25 m (Fig. 5) and the crown coverage was 96 %. FLiES-SIF assumes that all crowns have the same leaf area density. The spherical leaf angle distribution function was used. We also used three different landscape conditions, tropical broad leaf (Fig. forest (b)), Evergreen needleleaf (Fig. 5 (c)), and Savanna (Fig. 5 (d)) to understand the effect of landscape structure on SIF. The model requires optical data in the PAR domain and the spectral wavelength to be simulated. In this sensitivity analysis, we used the data assembled by Kobayashi (2015a). Figure 6 shows the spectral leaf reflectance and transmittance and the woody/soil reflectance. The leaf reflectance and transmittance, woody reflectance, and soil reflectance were calculated from various broadleaf spectral data, medium reflective woody elements, and medium reflective soil surfaces in Kobayashi (2015b), respectively. All optical

data were averaged over 10 – nm intervals between 650 nm and 850 nm. The optical data in the PAR domain were computed

as the average from 400 – 700 nm (Table 2). The same woody reflectance data were used for both stem surface and branch materials. The fractions of SIF emission ($f_{ada}$, $f_{aba}$) were determined using the FluorMODleaf model (FluorMODgui V3.1) (Zarco-Tejada et al., 2006; Pedrós et al., 2010) (Fig. 7). To run FluorMODgui V3.1, we used the default biochemical parameters (leaf structure parameter $N = 1.5$, chlorophyll a+b content $C_{ab} = 33.0$ µ g cm$^{-1}$, water content $C_w = 0.025$ cm, dry matter content $C_m = 0.01$ g cm$^{-2}$, fluorescence quantum efficiency $F_i = 0.01$, leaf temperature $T = 20.0$°C, species temperature

dependence = 2 (beans), stoichiometry of PSII (photosystem II) to PSI reaction centers $Sto = 2.0$) under the downward spectral sky radiation data (direct transmittance in sun direction ($\tau_s$), FluorMOD30V23.MEP). The fractions of SIF emission were derived from the simulated leaf fluorescence output by normalizing the simulated leaf level SIF from the adaxial and abaxial sides. In this sensitivity analysis, we employed two types of leaf-level SIF yield $\phi_f$. The first type is a constant value of 0.01 throughout the whole APAR range from Sect. 3.2 – 3.5. This value is used to test the impact of forest structures (LAI) and sun

and observation geometries on SIF. The second type is an APAR-dependent value derived using the models of Van der Tol et al. (2014) (Fig. 8) and Farquhar et al. (1980). Tol's model is based on energy partition within leaves in Sect. 3.6. In calculating $\phi_f$, we used the leaf fluorescence and physiology module in the FLiES-SIF model as described in Sect. 2.1. The parameter values in these models were set by reference to previous literature (such as Van der Tol et al. (2014) and De Pury and Farquhar (1997)), and the results compared with those using a constant APAR-dependent (Tol's model) $\phi_f$. In the test simulation except

Sect. 3.5, the incident total PAR on the canopy surface was fixed at 2000 µ mol m$^{-2}$ s$^{-1}$, except for in the APAR sensitivity analysis, and the fraction of diffuse radiation was fixed at 0.3. In the sensitivity analysis, we used $10^5 – 10^6$ photons in each model run. Figure 9 indicates the dependency of SZA and LAI on total SIF radiance between 650 to 850 nm. In the following section, we analyze the SIF sensitivity in $\lambda = 760$ nm.

## 3.2 Intercomparisons with the DART model

To demonstrate the efficacy of FLiES-SIF for calculating the SIF radiation, we compared the output with that of DART (Gastellu-Etchegorry et al., 2017), which is one of the available 3D models. We adopted the flux-tracking mode in the radiative method of DART ver. 5.7.6 to calculate the SIF radiation, and compared the simulation results under the following conditions: SZA = 30°, SAA (Solar Azimuth Angle) = 0°, and $\lambda = 760$ nm. Only part of the default landscape (40 m × 40 m, 50 trees) was simulated to reduce the computational load. Figure 10 compares the SIF radiation calculated by FLiES-SIF with

that given by DART-SIF. In terms of VZA dependency, FLiES-SIF overestimates SIF radiation by about 18 % on average in the forward direction (VZA > 0), and the difference becomes larger as VZA increases. In contrast, FLiES-SIF underestimates the SIF radiation by about 12 % in the backward direction (VZA < 0), and the difference reaches a maximum at VZA = −50°. In terms of LAI dependency, FLiES-SIF overestimates SIF radiation by about 9 % on average, with the difference being especially pronounced when LAI > 6. The SIF radiation increases with LAI in FLiES-SIF, but decreases as LAI increases in the

DART model. As a result, FLiES-SIF gives similar SIF radiation values as DART, and has greater sensitivity to angles.

FLiES-SIF is a reasonable and proper model for determining the SIF radiation. The DART model has a useful GUI and can calculate SIF radiation on complex landscape structures using many kinds of 3D objects. However, unlike FLiES-SIF, DART

does not include a leaf physiological module. Additionally, FLiES-SIF requires less computational resources than the DART model.

## 3.3 Sensitivity analysis with default forest landscape

### 3.3.1 Angular dependency of SIF

Figure 9(a) shows the total SIF radiance for wavelengths between $650 - 850$ nm. These figures indicate that the SIF radiance shows a strong peak near the sun direction over the whole wavelength range, although the SZA value, which exhibits the maximum SIF, varies according to the wavelength. In the visible red region, the SIF radiance reaches an extremum at a lower SZA than in the near-infrared region. Regardless of wavelength, the angular dependency of SIF exhibits similar patterns: in the direction of forward emission (VZA > 0), SIF increases with an increase in VZA and sharp strong peaks appear around the sun direction (the hotspot effect). In the backward direction (VZA < 0), the SIF decreases with an increase in |VZA| and attains a minimum at around $-35° - -50°$, before increasing with |VZA|. Although the general angular patterns are similar across the whole wavelength range, the strength of the hotspot peak in the forward direction and the minimum SIF in the backward direction vary slightly with the wavelength.

To analyze the dependency of VZA in more detail, we explored the influence of VZA on three terms in Eq. (2), namely the direct SIF from the radiance of sunlit and shaded leaves ($I_{\text{dir\_sun}}$, $I_{\text{dir\_shade}}$) and the scattered radiance ($I_{\text{ms\_sun}} + I_{\text{ms\_shade}}$), as well as the total SIF radiance ($I$). The simulated SIF shows distinct angular features for each SIF component ($I_{\text{dir\_sun}}$, $I_{\text{dir\_shade}}$, $I_{\text{ms\_sun}} + I_{\text{ms\_shade}}$). Figure 11 shows the dependence on VZA of the SIF components when LAI = 3.0 for a wavelength of 760 nm. $I_{\text{dir\_sun}}$ has a strong peak near the sun direction because of the hotspot effect, whereas angular changes of $I_{\text{dir\_sun}}$ in other domains are minor. In contrast, $I_{\text{ms\_sun}} + I_{\text{ms\_shade}}$ exhibits bowl-like shapes (Fig. 11(d)), which contributes to the enhancement of total SIF at higher angles. In the FLiES-SIF model framework, SIF radiance is computed by collecting the contribution factor (Eqs. (5) and (15)) from the attenuation function, weight of photons, and phase function. Among those factors, the drastic changes in the optical thickness of the attenuation function (Eqs. (2) – (10)) contributed the most to the hotspot in $I_{\text{sun\_dir}}$. The attenuation function displays a strong peak around the sun direction because of the hotspot parameter ($H$ in Eq. (9)). When $\alpha_j$ is sufficiently large and the hotspot effect is marginal, the attenuation function is determined by the forest structure (such as LAI and leaf angle density). Away from the sun direction, the SIF radiance gradually decreases or increases slightly. This angular feature (VZA) is influenced by the initial photon weight and phase function through the dependency on the leaf surface normal: the initial photon weight is calculated as the inner product between the leaf angle and the sun direction. The influence of SZA on the phase function is greater than that on the initial photon weight. The other two components ($I_{\text{dir\_shade}}$, $I_{\text{ms\_sun}} + I_{\text{ms\_shade}}$) contribute to the total SIF increase in higher angular domains. In addition, $I_{\text{dir\_shade}}$ makes a slightly larger contribution in the backward direction, because shaded leaves tend to be more aligned with the backward direction. The shaded leaves only absorb diffuse sky radiation, so the relative magnitude of $I_{\text{dir\_shade}}$ with respect to $I_{\text{dir\_sun}}$ greatly depends on the fraction of diffuse radiation. The contributions of these three components to the direct in four difference sun angles are presented in Fig. 12. These

445 partitions vary with the fraction of incoming diffuse radiation, optical properties (leaf reflectance and transmittance, woody and soil reflectance), and the leaf area.

### 3.3.2 Angular dependencies of APAR and sunlit leaves

Because SIF radiance is greatly affected by the APAR of the leaves, the angular behavior of APAR is essential in understanding the numerical computation of SIF emissions. In the FLiES-SIF model, the SIF radiance is first computed under the apparent
APAR ($APAR_{app}$) conditions (Sect. 2.2), and then adjusted by multiplying by the ratio of $APAR_c$ to $APAR_{app}$ (Eq. (3)). The simulated angular patterns indicate that $APAR_c$ increases with an increase in SZA (Fig. 13(b)). The increase in $APAR_c$ with respect to SZA corresponds to the increase in the photon pathlength inside the forest canopy. As SZA increases, more photons are likely to hit leaves before they pass through the canopy layers. In contrast, $APAR_{app}$ decreases as SZA decreases (Fig. 13(a)). This is because $APAR_{app}$ is related to the fraction of sunlit leaves. As described in simulation procedure C in
Sect. 2.5, the photon tracing is initiated from either sunlit or shaded leaves at randomly selected positions. As the LAI along the photon path ($L_p$) increases, the gap fraction $P_{sun}$ becomes smaller (Eq. (17)). As a result, shaded leaves are more likely to be selected in the random process in Eq. (18). In other words, as the fraction of shaded leaves increases, the amount of energy in the simulated system decreases. In the Monte Carlo simulations, the statistical accuracy of the simulated variables depends on the number of photon samplings. The decrease in $APAR_{app}$ does not affect the simulated accuracy of the total SIF radiance;
however, it does affect the individual components in Eq. (2), which means the statistical accuracy of $I_{dir\_sun}$ decreases as SZA increases. Depending on the target sampling variables to be simulated, the number of photons should be determined (i.e., more photons may be necessary to investigate the behavior of $I_{dir\_sun}$ in cases where the sunlit leaf fraction is low).

### 3.3.3 Leaf area density dependency

Figure 9(b) shows the sensitivity of total SIF radiance to LAI for wavelengths of 650 – 850 nm. The simulated SIF increases
with LAI and then becomes saturated over the whole wavelength range, although the speed of saturation varies with the wavelength. In the visible domain, the simulated SIF becomes saturated when LAI = 2. In the near-infrared domain, the simulated SIF is not saturated at higher LAI values, indicating that SIF is more sensitive to LAI in the near-infrared domain. To analyze the dependency on LAI, we explored the influence of LAI on three terms in Eq. (2), namely $I_{dir\_sun}$, $I_{dir\_shade}$, and $I_{ms\_sun} + I_{ms\_shade}$, as well as the total SIF radiance ($I$) (forward direction in Fig. 14 and backward direction in Fig. 15). In
our simulation scenarios, $I_{dir\_sun}$ contributed about 54 % of total SIF radiance when LAI = 3, VZA = 10°, and SZA = 20°. $I_{dir\_hade}$ and $I_{ms\_sun} + I_{ms\_shade}$ contributed 7 % and 39 %, respectively (Fig. 16). Figures 14 and 15 show that the individual SIF components respond differently to the LAI.

**i. Direct SIF radiance from sunlit leaves**

The LAI dependency of direct SIF radiance from sunlit leaves is influenced by the hotspot function and the magnitude of VZA
(Figs. 14(b) and 15(b)). Generally, the SIF radiance emitted from sunlit leaves increases and then saturates as LAI increases,

because the number of sunlit leaves also increases and becomes saturated, although the fraction of sunlit leaves decreases (Fig. 17(c)). However, in terms of simulated SIF radiance, there are ranges of LAI in which SIF radiance decreases with an increase in LAI. In these regions, the decrease in SIF radiance is caused by the attenuation of SIF radiance in the canopy. The magnitude of this attenuation depends on both the hotspot function and VZA. The hotspot function (i.e., the angle $\alpha_j$ in Eq. 9) has a major influence on simulated direct SIF radiance from sunlit leaves. The SIF radiance increases and then becomes saturated without decreasing when $\alpha_j$ is equal to 0, because the rate of decrease in $I'_{\mathrm{dir\_sun}}$ becomes small when $\tau = 0$. Additionally, smaller values of $\alpha_j$ produce a smaller rate of decrease in $I'_{\mathrm{dir\_sun}}$ with respect to increases in LAI through the hotspot effect. The magnitude of VZA (i.e., |VZA|) also influences the simulated SIF radiance. Generally, larger LAI values lead to a decrease in the attenuation of SIF radiation from sunlit leaves in the canopy when VZA is positive, because most sunlit leaves inhabit the canopy surface. However, the attenuation of SIF radiation in other canopies increases with |VZA| because of the increase in the pathlength to the canopy boundary when passing through other canopies. The influences of |VZA| and LAI are prominent in negative VZA directions. In this case, the decrease in SIF radiance with an increase in LAI becomes significant because of the SIF emitted through the local canopy to the view point, and the attenuation in the local canopy (and in other canopies) increases with LAI. Thus, the increase in pathlength as |VZA| increases significantly affects $I'_{\mathrm{dir\_sun}}$ in the view direction.

### ii. Direct SIF radiance from shaded leaves

The fraction of shaded leaves has a major influence on SIF radiance. SIF increases and then becomes saturated without decreasing when VZA is negative (Figs. 14(c) and 15(c)). This variation in SIF is caused by an increase in the fraction of shaded leaves, because the rate of increase in the fraction is larger than the rate of decrease in $I'_{\mathrm{dir\_shade}}$. In contrast, the rate of decrease in $I'_{\mathrm{dir\_shade}}$ becomes greater than the rate of increase in the fraction of shaded leaves when VZA is positive. In this region, the expectation of the pathlength to the view point is larger than for negative VZA, because the canopy surface is covered with sunlit leaves. This increase in optical thickness, which depends on the pathlength, has a major effect on $I'_{\mathrm{dir\_shade}}$ in the LAI range where $\psi$ rapidly decreases with any increase in $\tau'$.

### iii. Scattered SIF radiance

The scattered SIF radiance refers to the sum of the scattered radiance from sunlit and shaded leaves, $I'_{\mathrm{ms}} \left( = I'_{\mathrm{ms\_sun}} + I'_{\mathrm{ms\_shade}} \right)$ in our model. The LAI dependency with respect to view direction on the scattered SIF radiance is in contrast to the direct radiance from shaded leaves (Figs. 14(d) and 15(d)). When VZA is positive, the SIF radiance increases and then becomes saturated without decreasing. The pathlength from sunlit leaves to the population boundary in the view direction has a major influence on simulated scattered SIF radiance. As previously explained (Sect. 3.3.3), the surface of the canopy is covered by sunlit leaves, which provide a large photon weight to scattered photons, in the positive VZA direction. When LAI is large, the decrease in $I'_{\mathrm{ms}}$ with an increase in LAI becomes vanishingly small. This is because the scattered radiation from high-weight photons reaches the view point with little attenuation. Larger values of LAI lead to shorter scattering pathlengths and fewer

scatterings, so the photon weight $w_j$ is larger. Additionally, the pathlength between sunlit leaves and the boundary of the canopy is nearly constant, irrespective of LAI variation.

The simulated SIF radiance, therefore, becomes larger than the radiance in the negative VZA direction. Actually, the expectation of the product of $w$ and $exp(-\tau')$ is larger than when VZA is negative (Fig. 15). In contrast, with an increase in LAI, the SIF radiance decreases and becomes saturated after increasing because of the increase in $\tau'$ from sunlit leaves. This is for a similar reason as for $I'_{\mathrm{dir\_shade}}$ when SZA is negative.

    Figure 17 shows APAR and the fraction of sunlit leaves as a function of LAI. $\mathrm{APAR_c}$ increases with an increase in LAI

and becomes saturated at around LAI = 2. $\mathrm{APAR_{app}}$ and the fraction of sunlit leaves decrease when LAI < 2. The increase in $\mathrm{APAR_c}$ and the decrease in $\mathrm{APAR_{app}}$ are more abrupt than the SIF increase with respect to LAI. This is because APAR is the visible light where the absorption of green leaves is high (~0.9). Thus, the $\mathrm{APAR_c}$ saturation curve has similar patterns of visible SIF radiance. At higher LAI, the fraction of sunlit leaves is low and $\mathrm{APAR_{app}}$ decreases. The statistical accuracy of $I'_{\mathrm{dir\_sun}}$ becomes drastically lower as $\mathrm{APAR_{app}}$ and the fraction of sunlit leaves decrease. Accurate simulations of $I'_{\mathrm{dir\_sun}}$

require an increased number of photons to be traced.

### 3.4   Variation of landscape

FLiES-SIF is applicable to a wide variety of landscapes. Figure 18 shows the LAI and SZA dependencies of the total SIF radiation in three different landscapes, namely tropical broadleaf forest, evergreen needle forest, and savanna, which have different tree densities, shapes, and crown coverages (Yang et al., 2018). These simulations use same optical data to demonstrate

the applicability of our model. The SIF radiation increases with LAI and then becomes saturated, and there is little difference in LAI dependency among these landscapes (upper panels in Fig. 18). In contrast, the angular dependency is affected by the landscape composition. For example, in the case of the savanna (Fig. 18(c)), the hotspot peak is smaller than for the other landscape types because the attenuation within other crowns is reduced by the lower crown coverage.

### 3.5   Comparison of 1D and 3D actual diurnal variations in PAR

Simulations with a one-hour step size were performed using observed total and diffuse PAR data from Yokohama, Japan from July 12 – 18 (DOY 192 – 198), 2014 (Fig. 19(a)). This period included clear sky days (DOY 192, 195, and 196) and overcast days (DOY 193, 198). The LAI value was fixed to 3.0 throughout the simulation period in all scenarios. The diurnal variations in sunlit LAI show distinct features with respect to 1D and 3D landscapes, as well as to the leaf angle distribution (Fig. 19(b)). In summary, the simulated sunlit LAIs in the 1D scenarios (1DsphNC, 1DereNC, 1DplaNC) are higher than those in the 3D

scenarios (3DsphNC, 3DereNC, 3DplaNC, 3DsphWC), and the difference in the leaf angle distribution affects the diurnal feature of sunlit LAI significantly: erectrophile cases give the highest values, followed by spherical and then planophile. The planophile cases (1DplaNC, 3DplaNC) exhibit weak diurnal variations because of the high probability of horizontal leaves: the uppermost leaves receive most of the incoming light, regardless of SZA. The sunlit LAI of the within-crown clumping scenario (3DsphWC) is significantly lower than in the no-clumping scenario (3DsphNC, 29.7 % lower at noon). Figures 19(c) and 19(d)

show the hourly simulation results for the top-of-canopy SIF at 760 nm.

The simulated SIF generally follows the diurnal variations in incoming PAR; however, the absolute range varies greatly depending on the leaf angle distribution (spherical, erectrophile, or planophile) and in the 1D and 3D cases. For example, the canopy SIF simulated by 1DplaNC is 2.3 times higher than that of 3DereNC at noon on a clear day (DOY 192). In contrast, the simulated SIF differences are minor on overcast days (DOY 193, 198). The simulated results also indicate that there is a sort of "trade-off" relationship between the amount of sunlit LAI and SIF: the SIF values simulated by planophile scenarios (1DplaNC, 3DplaNC) are higher than those in erectrophile scenarios (1DereNC, 3DereNC), while the amount of sunlit LAI in planophile scenarios is lower than that in erectrophile scenarios. This is because leaves with an erectrophile distribution receive the solar beam efficiently as they have a large proportion of vertical leaves. This, in turn, results in larger incident angles of the solar beam on the leaf surfaces, which reduces the incident PAR intensity on the unit leaf area. Comparing the 3D scenarios with the 1D scenarios, the latter simulated the SIF to be 33 – 41 % higher than the former.

Additionally, we investigated the contribution of direct SIF from sunlit leaves ($I_{\mathrm{dir\_sun}}$), shaded leaves ($I_{\mathrm{dir\_shade}}$), and multiple scattering components ($I_{\mathrm{ms\_sun}} + I_{\mathrm{ms\_shade}}$) (Figs. 19(e) and 19(f)). The largest contribution comes from sunlit leaves. The contribution from shaded leaves is generally lower than that from sunlit leaves, although on overcast days (DOY 193 and 198) the contribution becomes close to or even higher than that of the sunlit leaves (Figs. 19(e) and 19(f)). Multiple scattering contributes substantially in the near-infrared domain (30 – 40 % of total SIF radiance), but is expected to make a lower contribution in the red spectrum because of low leaf reflectance and transmittance. A comparison of 3DsphNC and 3DsphWC shows that the shaded leaves of the within-crown clumping scenario contribute more to the total SIF than in the no-clumping scenario. Overall, the 1D and 3D comparisons, as well as differences in leaf angle distributions, suggest that reasonable assumptions must be made regarding the canopy structure if the SIF simulations are to be reliable. If 1D models are applied to forest canopies, the simulated SIF is prone to be overestimated.

## 3.6  Influence of fluorescence yield on variable $\mathrm{APAR_L}$ scenario

Figure 20 compares the total SIF derived from fixed and variable leaf-level SIF yields. To explore the influence of the SIF yield on the above dependencies, we derive $\phi_{\mathrm{f}}$ by means of Tol's model (Van der Tol et al., 2014) to calculate the yield from APAR, which is obtained by our model (Fig. 8). Additionally, we obtain the photosynthesis yield by Farquhar's model using parameter values set by reference of previous literatures (such as Van der Tol et al. (2014) and De Pury and Farquhar (1997)) to derive the $\phi_{\mathrm{f}}$ without observation data. Figure 20 compares the total SIF radiance $I$ between models based on constant $\phi_{\mathrm{f}}$ and APAR-dependent $\phi_{\mathrm{f}}$ in terms of their dependence on LAI and SZA, respectively. The dependency on the two parameters is not substantially different because the variation in $\phi_{\mathrm{f}}$ is smaller than that of APAR. However, $\phi_{\mathrm{f}}$ affects $\mathrm{APAR_{app}}$ as well as the bidirectional SIF radiance. Thus, obtaining accurate values of $\phi_{\mathrm{f}}$ is important in estimating the exact level of SIF; this issue will be considered in future work.

## 4 Conclusions

In this paper, we have described the structure of FLiES-SIF version 1.0 and the simulation algorithm for canopy-scale sun-induced chlorophyll fluorescence emissions. The model was developed by extending the original FLiES model. FLiES-SIF is based on the Monte Carlo ray tracing approach. The SIF emissions from sunlit and shaded leaves are computed separately, and the model also considers multiple scatterings within forest canopies. FLiES-SIF version 1.0 simulates virtual forest landscapes, where individual tree positions and crown dimensions are explicitly considered. Therefore, the model can examine the influence of various ecological and environmental factors (e.g., forest structures and solar direction) on SIF emissions in a realistic canopy. A 3D radiative transfer modeling approach is necessary for understanding the biological and physical mechanisms behind the SIF emissions from complex forest canopies. We performed a test run to demonstrate the sensitivity of SIF to the view angle, LAI, and leaf-level SIF yield. The simulation results show that SIF increases with LAI before becoming saturated when LAI > 2 – 4, depending on the spectral wavelength. The sensitivity analyses also showed that simulated SIF radiation may decrease with LAI when LAI > 5. These phenomena were observed under certain sun and view angle conditions. This type of nonlinear and nonmonotonic SIF behavior is also related to the spatial forest structure patterns and leaf angle distributions. The simulated SIF with a 1D canopy assumption is prone to be overestimated. The hotspot effect plays an important role in SIF simulations when the view direction is close to the sun direction. The SIF yield $\phi_f$ influences the canopy SIF, especially when APAR is low. FLiES-SIF version 1.0 can be used to quantify the canopy SIF at various view angles, including the contribution of multiple scatterings, which is an important component in the near-infrared domain. The proposed model can be used to standardize satellite SIF by correcting the bidirectional effect. This step will contribute to improved GPP estimation accuracy through SIF. In this model description paper, we have focused on the formulation and simulation schemes of FLiES-SIF version 1.0, and have presented the results from sensitivity analyses of major variables such as LAI. Model validation using field measurements will be performed in future studies. Thorough validation against measured quantities should be conducted to evaluate the accuracy of the model.

*Code availability.* The FLiES-SIF version 1.0 source code and sample data sets used in this study are publicly available through Zenodo (Kobayashi and Sakai, 2019, http://doi.org/10.5281/zenodo.3584099). The source codes are written in Fortran (gfortran), C (gcc) and R script. This is an open source software under the agreement of Creative Commons Attribution 4.0 International license (CC BY 4.0).

*Author contributions.* YS, HK and TK designed the FLiES-SIF version 1.0 model. YS and HK developed the FLiES-SIF version 1.0. YS carried out the sensitivity analysis. YS and HK wrote the paper.

*Competing interests.* The authors declare that they have no conflict of interest.

*Acknowledgements.* We thank Dr. Jean-Philippe Gastellu-Etchegorry for his kind support and instructive advice in running the DART-SIF module. The FLiES-SIF version 1.0 development was supported by the Environment Research and Technology Development Fund (2RF-1601, 2-1903) and JSPS Kakenhi (25281014, 16H02948, 18H03350). We thank Stuart Jenkinson, PhD, from Edanz Group (https://en-author-services.edanzgroup.com/) for english grammatical editing a draft of this manuscript.

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

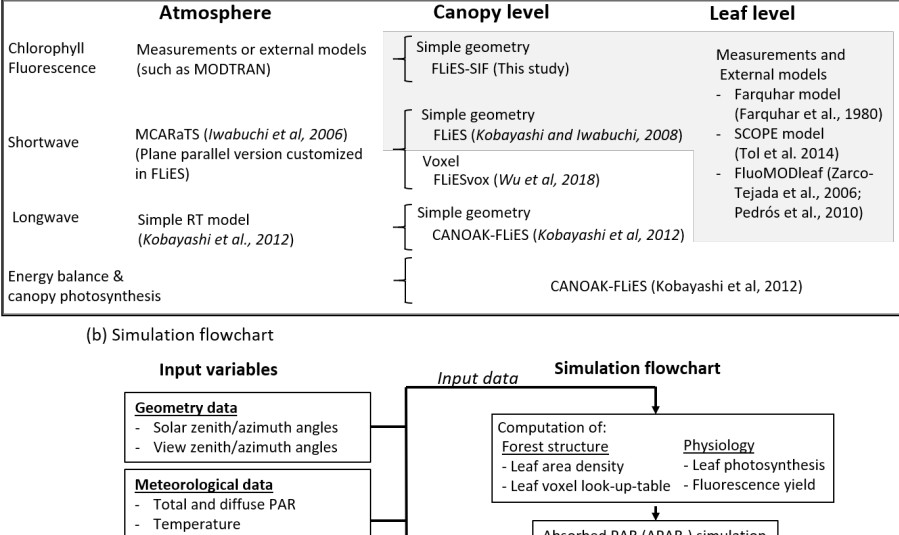

(a) General framework of the FLiES modules

|  | Atmosphere | Canopy level | Leaf level |
|---|---|---|---|
| Chlorophyll Fluorescence | Measurements or external models (such as MODTRAN) | Simple geometry FLiES-SIF (This study) | Measurements and External models<br>- Farquhar model (Farquhar et al., 1980)<br>- SCOPE model (Tol et al. 2014)<br>- FluoMODleaf (Zarco-Tejada et al., 2006; Pedrós et al., 2010) |
| Shortwave | MCARaTS (Iwabuchi et al, 2006) (Plane parallel version customized in FLiES) | Simple geometry FLiES (Kobayashi and Iwabuchi, 2008)<br>Voxel FLiESvox (Wu et al, 2018) | |
| Longwave | Simple RT model (Kobayashi et al., 2012) | Simple geometry CANOAK-FLiES (Kobayashi et al., 2012) | |
| Energy balance & canopy photosynthesis | | CANOAK-FLiES (Kobayashi et al, 2012) | |

(b) Simulation flowchart

**Figure 1.** General outline of the FLiES modules and simulation flows. (a) General framework of the FLiES modules. Newly developed module in the current study is indicated by the gray color background. (b) Simulation flow and input data sets. Four major input data, geometry data, meteorological data, forest stand data, and optical and leaf trait data, are required to run the model.

**Table 1.** Simulation scenarios for the 1D and 3D comparisons with an actual diurnal incident PAR data.

| Scenario ID | Model dimension | LAD functions | clumping |
|---|---|---|---|
| 1DsphNC | 1D | spherical | No |
| 1DereNC | 1D | erectrophile | No |
| 1DplaNC | 1D | planophile | No |
| 3DsphNC | 3D | spherical | No |
| 3DereNC | 3D | erectrophile | No |
| 3DplaNC | 3D | planophile | No |
| 3DsphWC | 3D | spherical | Yes ($\gamma = 0.6$) |

LAD: Leaf angle distribution

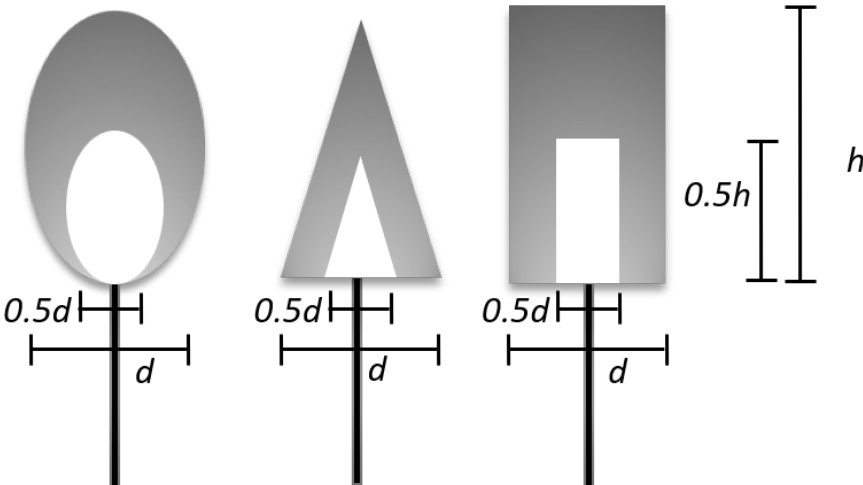

**Figure 2.** Representation of the individual crowns and stems. The tree crown objects are defined as either cone, cylinder, or spheroid, where d is the maximum diameter of the object and h is a crown height. The crown objects are divided by two domains. Outer domains (grey colored domains in the figure) are filled with green leaves. Inner domains are filled with woody materials. In the default setting, the size of inner domains are set as half of the crown size.

**Table 2.** Optical data in PAR domain used in the sensitivity analysis

| Leaf reflectance | Leaf transmittance | Woody reflectance | Soil reflectance |
|---|---|---|---|
| 0.06814 | 0.04192 | 0.18895 | 0.12952 |

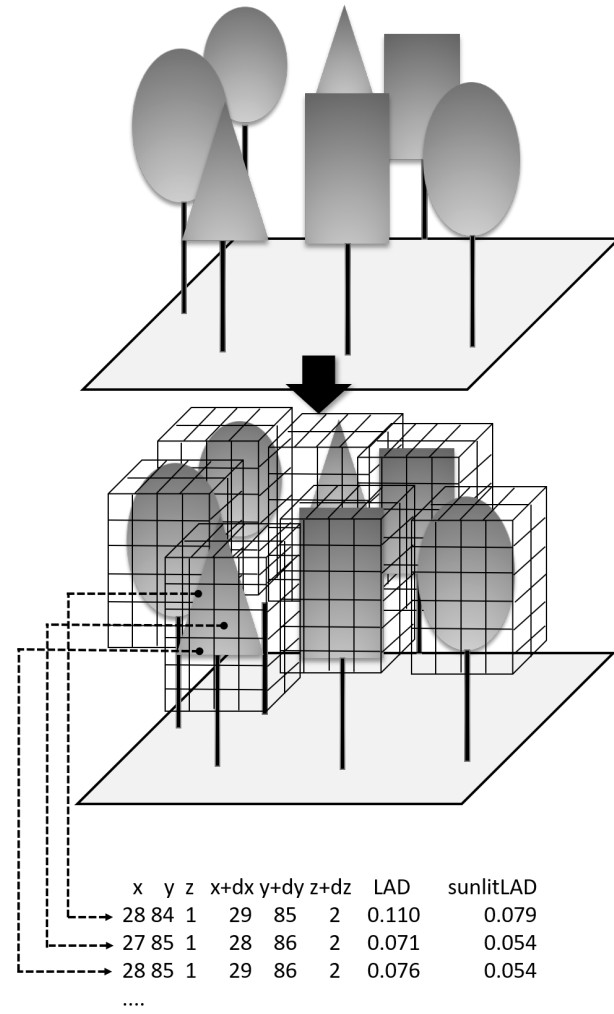

| x | y | z | x+dx | y+dy | z+dz | LAD | sunlitLAD |
|---|---|---|------|------|------|-----|-----------|
| 28 | 84 | 1 | 29 | 85 | 2 | 0.110 | 0.079 |
| 27 | 85 | 1 | 28 | 86 | 2 | 0.071 | 0.054 |
| 28 | 85 | 1 | 29 | 86 | 2 | 0.076 | 0.054 |
| .... | | | | | | | |

**Figure 3.** Voxel extraction from geometric canopy landscapes. The leaf voxel is extracted before the ray tracing simulation. FLiES-SIF model uses the geometric object approach for the Monte Carlo ray tracing. For the SIF simulation, the ray tracing is initiated from the randomly selected positions in the forest landscape. This voxel information is used to efficiently select the leaf position where SIF occurs.

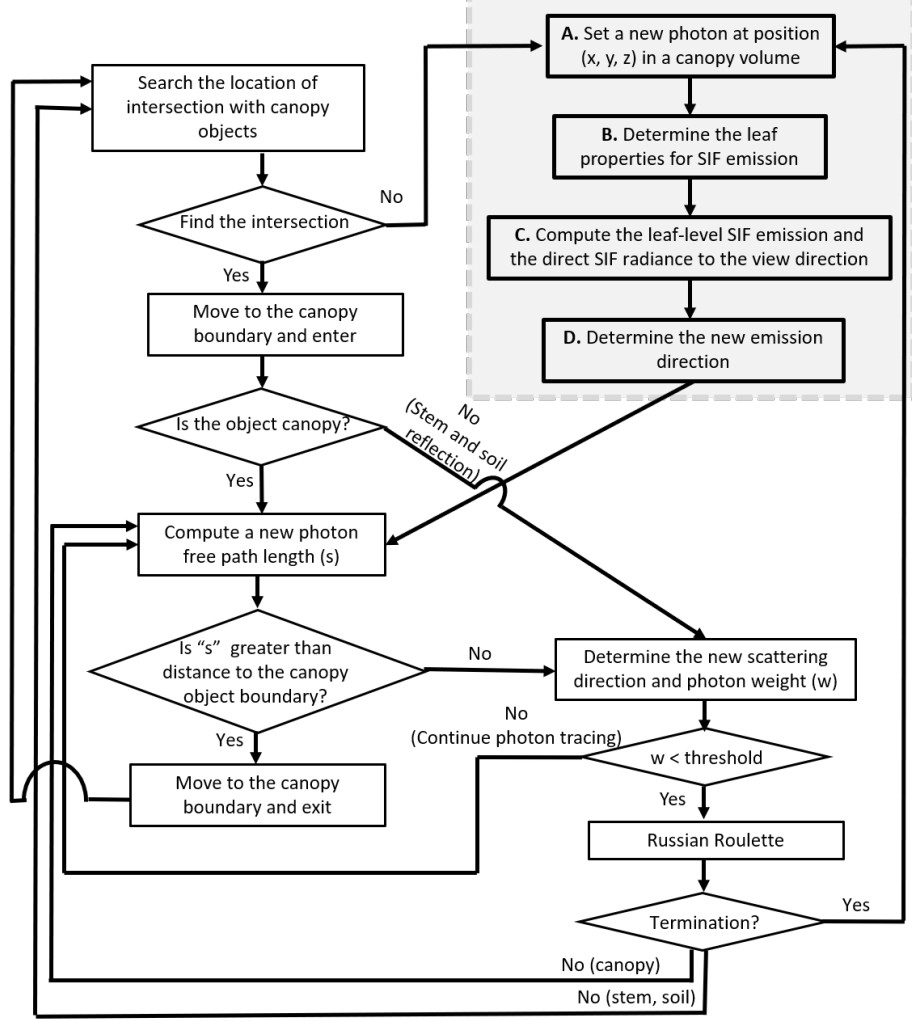

**Figure 4.** The flowchart of the Monte Carlo photon tracing scheme in canopy landscapes at a single-wavelength. The procedures A to E framed by the dotted grey rectangle indicate the photon tracing scheme for direct SIF emission. The other part of the flowchart corresponds the multiple scattering. The multiple scattering schemes are the same as the original FLiES model (Kobayashi and Iwabuchi, 2008).

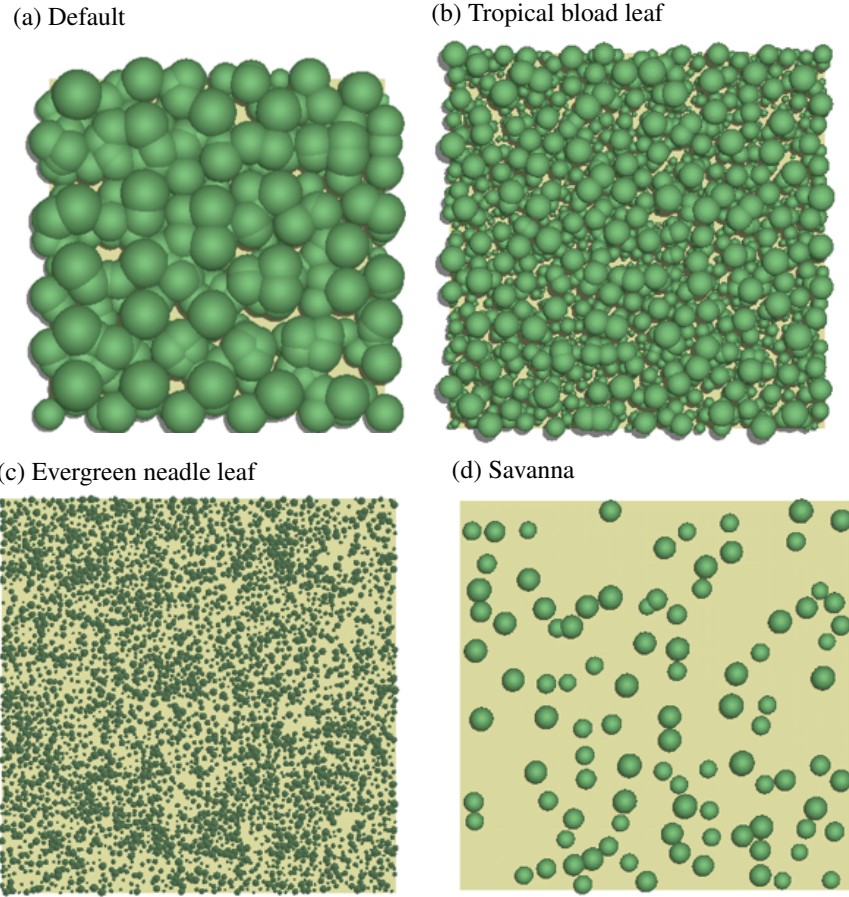

(a) Default

(b) Tropical bload leaf

(c) Evergreen neadle leaf

(d) Savanna

**Figure 5.** The forest landscape used in the sensitivity analysis. The landscape size is one-hectare (100 m × 100 m). The tree positions and canopy heights are determined by the random numbers. (a) Default forest: crown shape is spheroid, tree density is 359 tree $ha^{-1}$, canopy layer height is 5 − 20 m and crown coverage is 88 %. (b) Tropical broadleaf forest: crown shape is spheroid, tree density is 1816 tree $ha^{-1}$, canopy layer height is 5 − 30 m and crown coverage is 88 %. (c) Evergreen needle forest: crown shape is cone, tree density is 3592 tree $ha^{-1}$, canopy layer height is 5 − 15 m and crown coverage is 49 %. (d) Savanna: crown shape is spheroid, tree density is 96 tree $ha^{-1}$, canopy layer height is 5 − 10 m and crown coverage is 19 %.

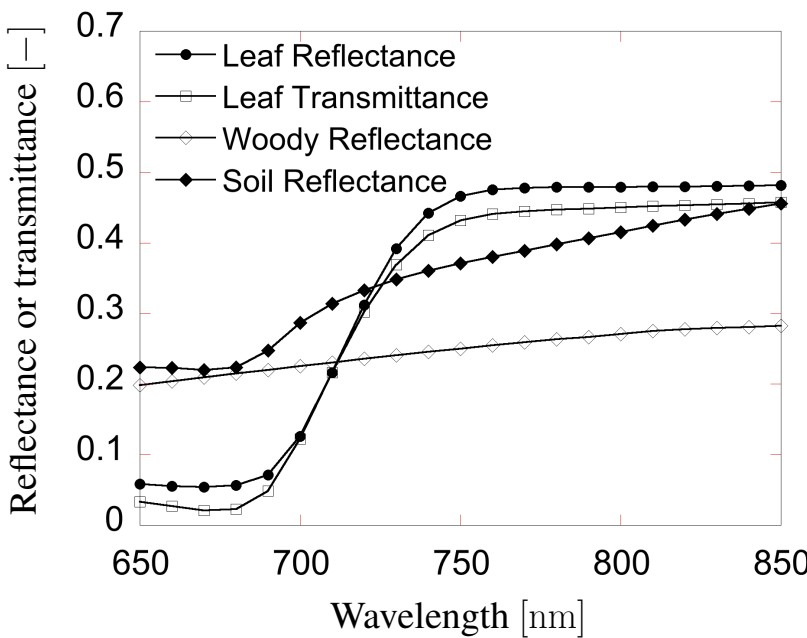

**Figure 6.** Spectral leaf reflectance and transmittance, woody reflectance and soil reflectance used in the sensitivity analysis. These optical data were constructed by averaging the spectral data in the literature and publicly available data sets (Kobayashi, 2015b).

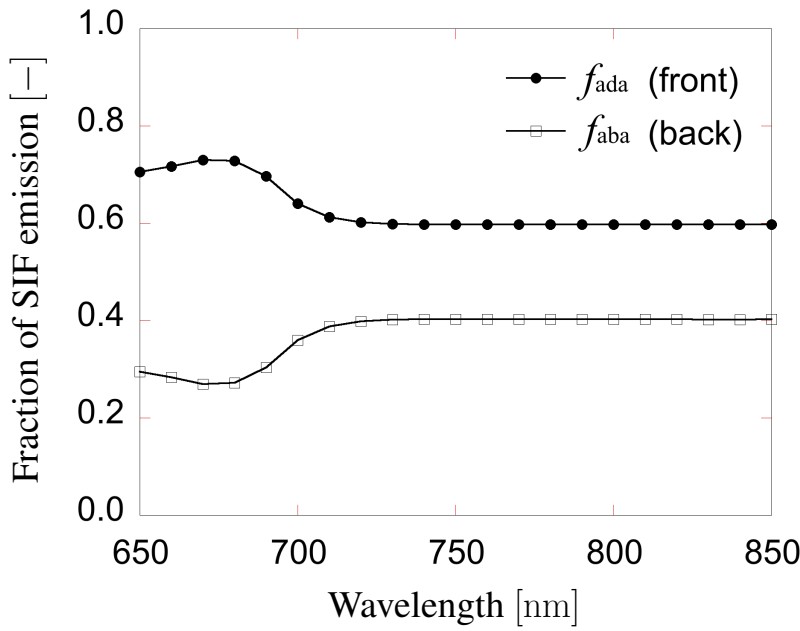

**Figure 7.** The fraction of SIF emission from adaxial and abaxial side of leaves. This ratio was determined by the leaf level chlorophyll fluorescence model (the FluorMODleaf model (FluorMODgui V3.1) (Zarco-Tejada et al., 2006; Pedrós et al., 2010).

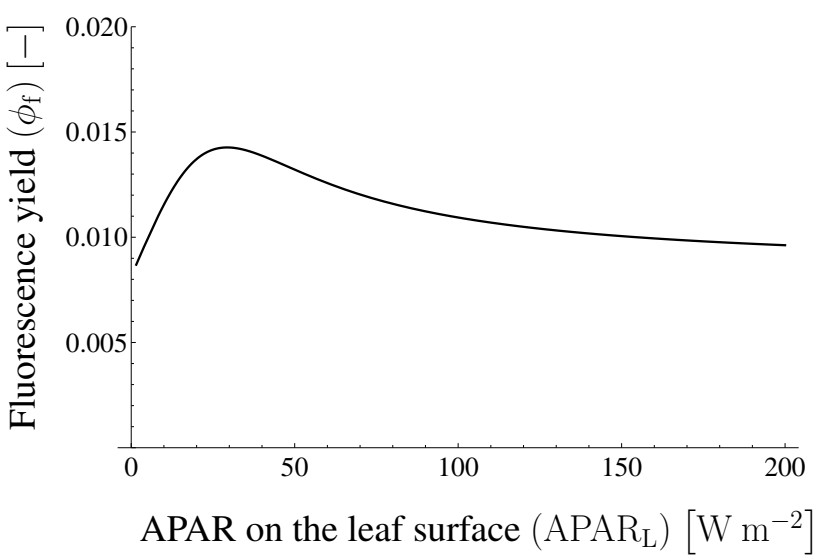

**Figure 8.** Fluorescence yield in Tol's model. The fluorescence yield depends on APAR on the leaf surface. In this case, $\phi_f$ is almost unchanged when $\mathrm{APAR_L}$ is greater than 200 W m$^{-2}$.

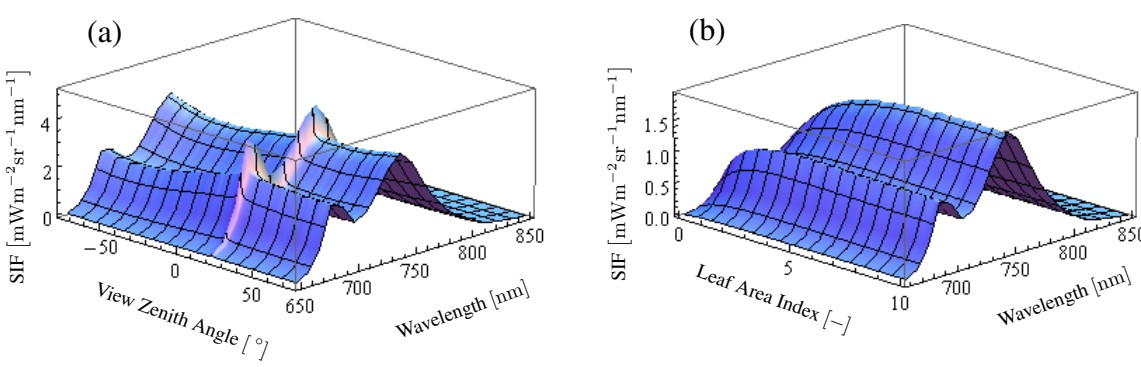

**Figure 9.** Variation of SIF radiance depending on VZA and LAI in wavelength 650 to 850 nm. Pannel (a) indicates VZA dependence of SIF at LAI = 3.0 and SZA = 20°. There is a strong peak in the sun direction at whole wavelength. Panel (b) indicates LAI dependence of SIF at VZA = 0° and SZA = 20°. SIF radiance increases with LAI and then becomes saturated.

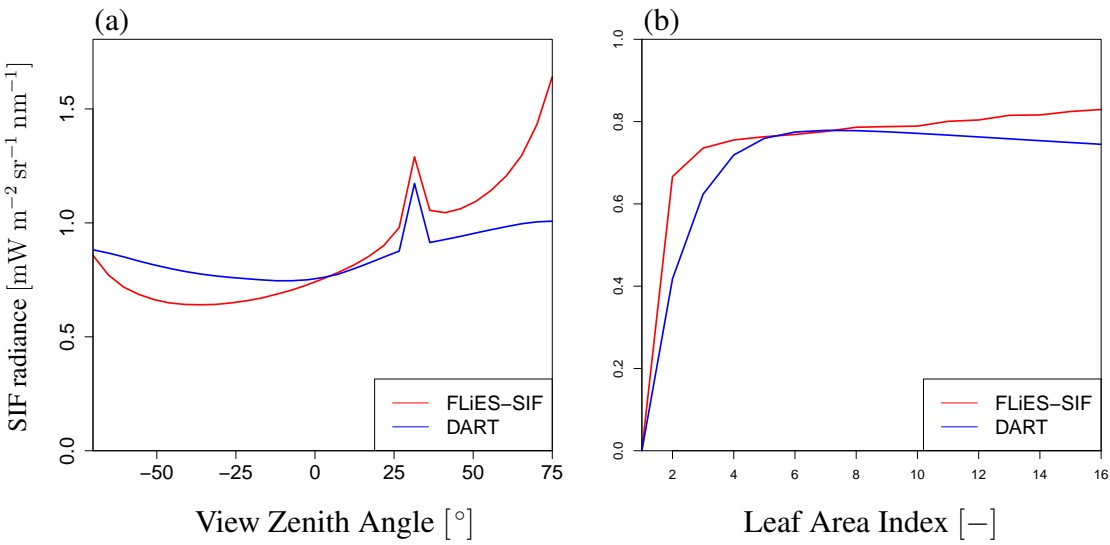

**Figure 10.** Comparison of SIF radiance with DART model. These figures indicate (a) VZA dependency (LAI = 3.0) and (b) LAI dependency (VZA = 0°), respectively. SZA and SAA value is 20° and 0°, respectively. The red and blue line indicates the result of FLiES-SIF and DART, respectively. FLiES-SIF has similar dependency on LAI and VZA to DART, although FLiES-SIF has higher angular dependency.

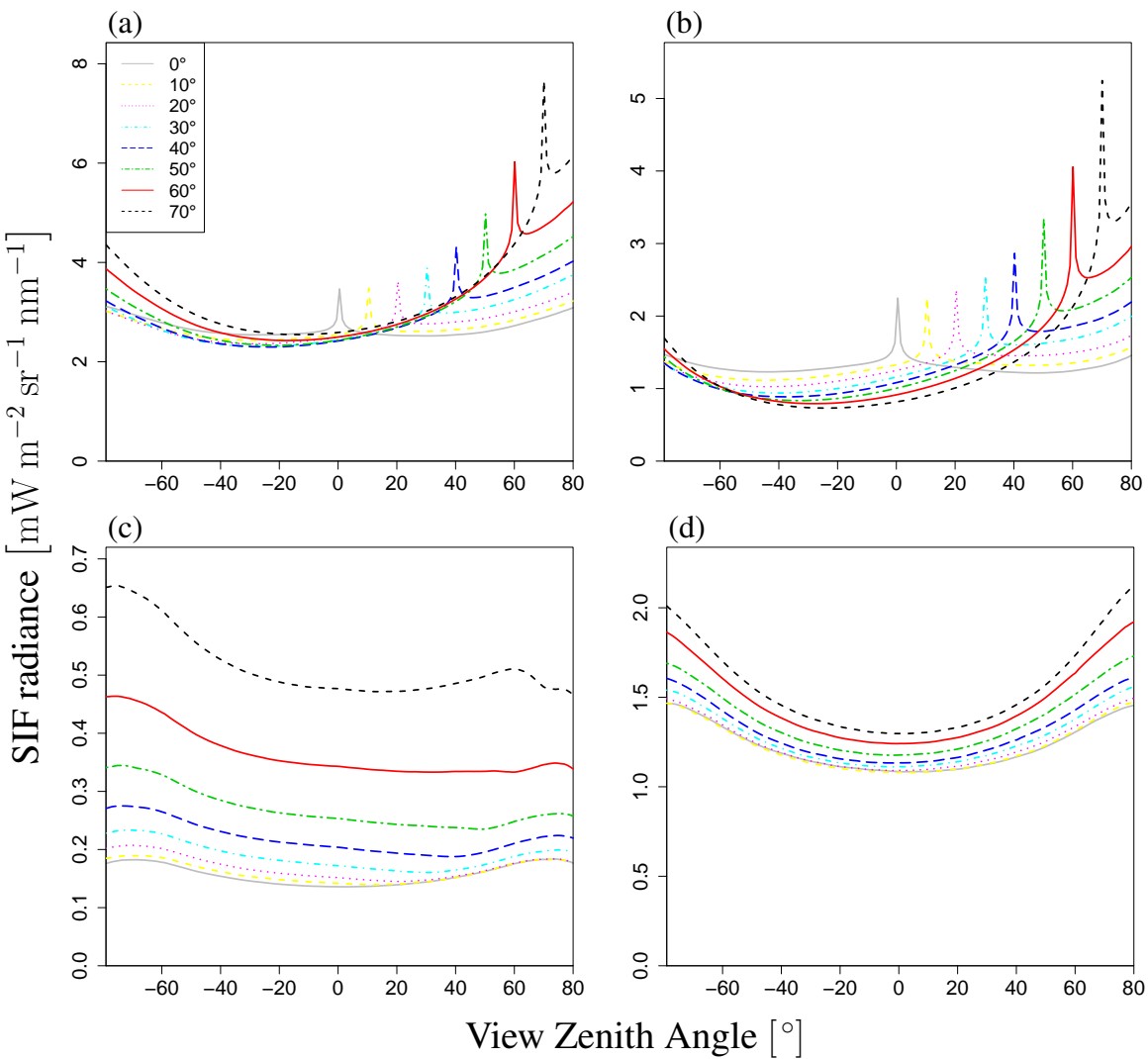

**Figure 11.** Angler dependence of SIF. This figure shows total radiance $I$ (a), direct radiance from sunlit leaves $I_{dir\_sun}$ (b), direct radiance from shaded leaves $I_{dir\_shade}$ (c), and radiance after multiple scatterings $I_{ms\_sun} + I_{ms\_shade}$ (d) at LAI = 3.0. Each line represents a different SZA (0° – 70°). Negative values of SZA represent the backward direction, and positive values represent the forward direction on the principal plane. The angular dependency varies greatly among these radiances.

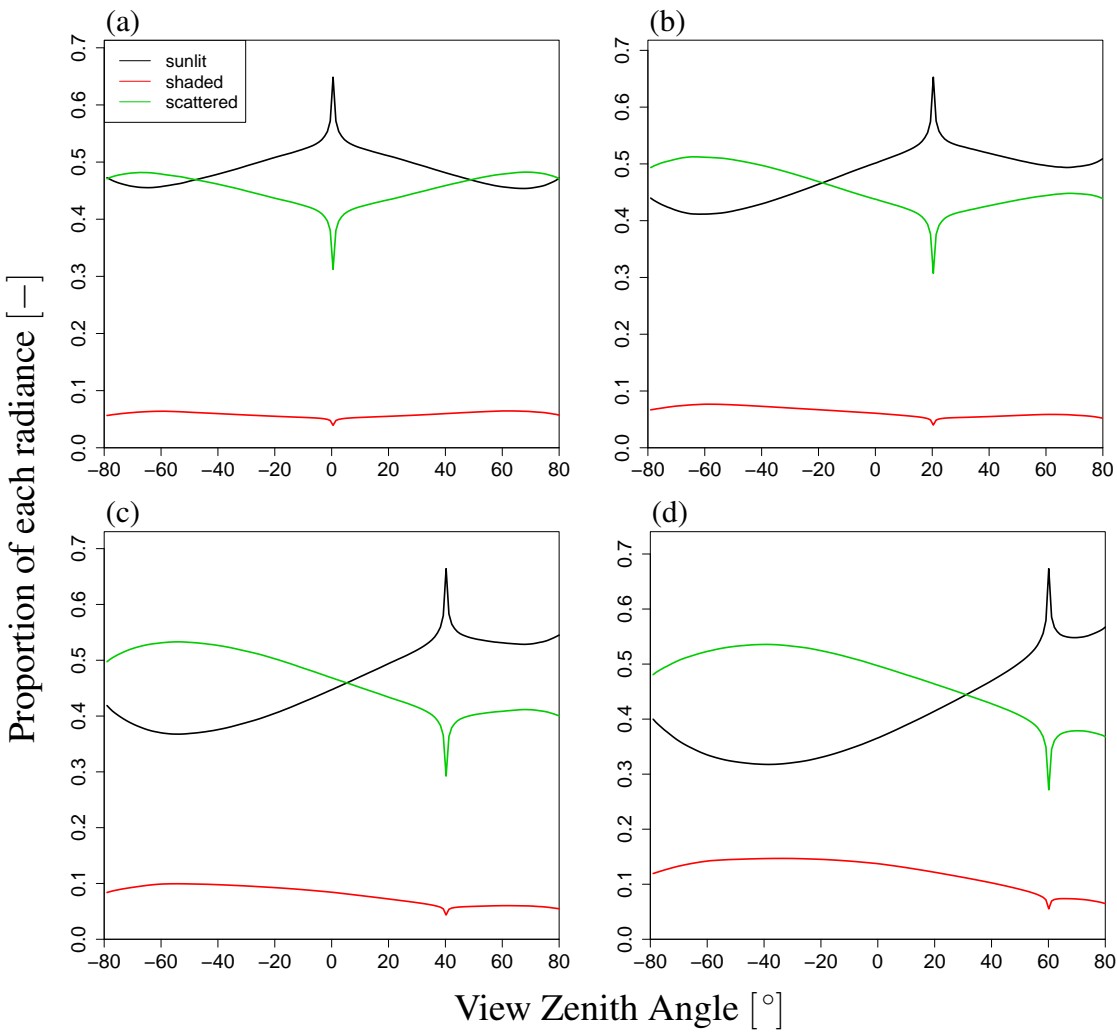

**Figure 12.** Proportion of SIF radiance with respect to VZA variation. Each figure shows the result under a different SZA value: (a) 0°, (b) 20°, (c) 40°, and (d) 60°. The contribution of shaded leaves is basically small and the contribution rates of the other two radiances exhibit some angular dependency. In the backward direction, the contribution of scattered radiation to SIF is greater than in the forward direction.

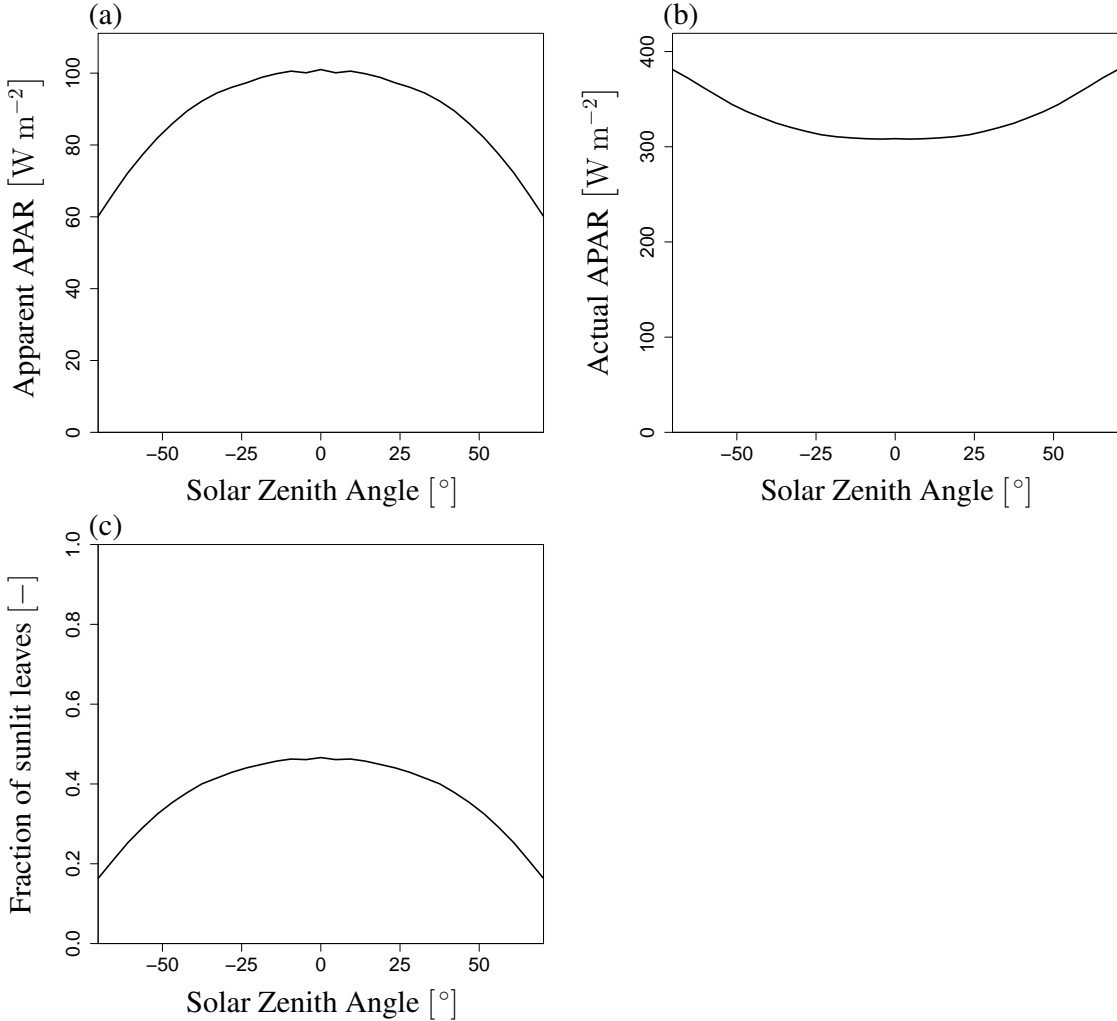

**Figure 13.** Variation of apparent and actual APAR and fraction of sunlit leaves with SZA. (a) Apparent APAR ($\text{APAR}_{\text{app}}$), (b) Actual APAR ($\text{APAR}_{\text{c}}$), and (c) Fraction of sunlit leaves ($\text{F}_{\text{sun}}$) at LAI = 3.0. These variables are not affected by VZA. $\text{APAR}_{\text{app}}$ and $\text{F}_{\text{sun}}$ decrease and $\text{APAR}_{\text{c}}$ increases with an increase in |SZA|.

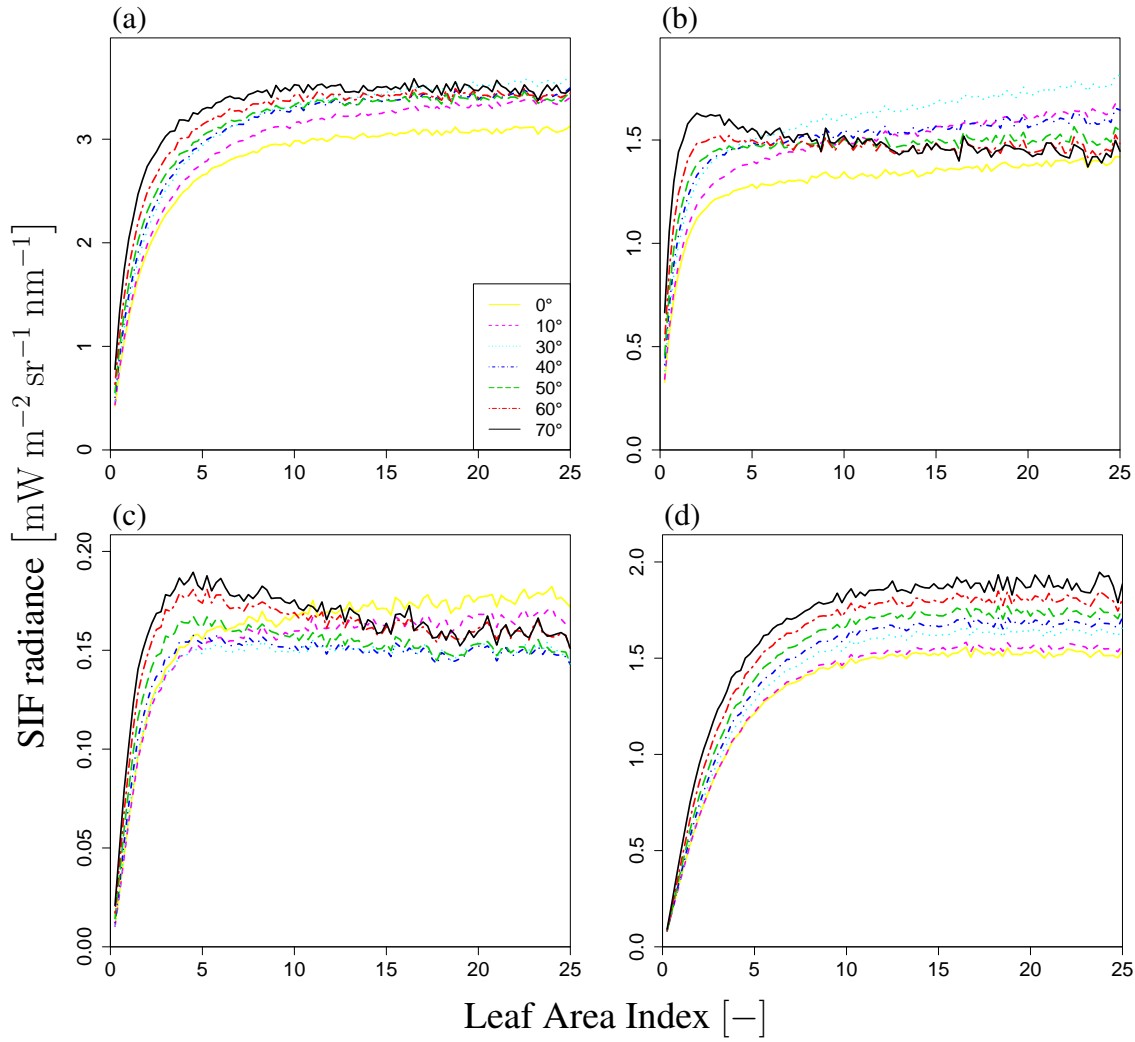

**Figure 14.** LAI dependency on SIF radiance. (a) Total radiance $I$, (b) direct radiance from sunlit leaves $I_{\text{dir\_sun}}$, (c) direct radiance from shaded leaves $I_{\text{dir\_shade}}$, and (d) radiance after multiple scatterings $I_{\text{ms\_sun}} + I_{\text{ms\_shade}}$ at SZA = 20° and SAA = 0° (forward direction). Each line represents a different VZA value (0° – 70°). SIF radiance increases with LAI and then becomes saturated in most cases. However, when VZA is large (e.g., black and red lines), the direct radiance decreases with an increase in LAI.

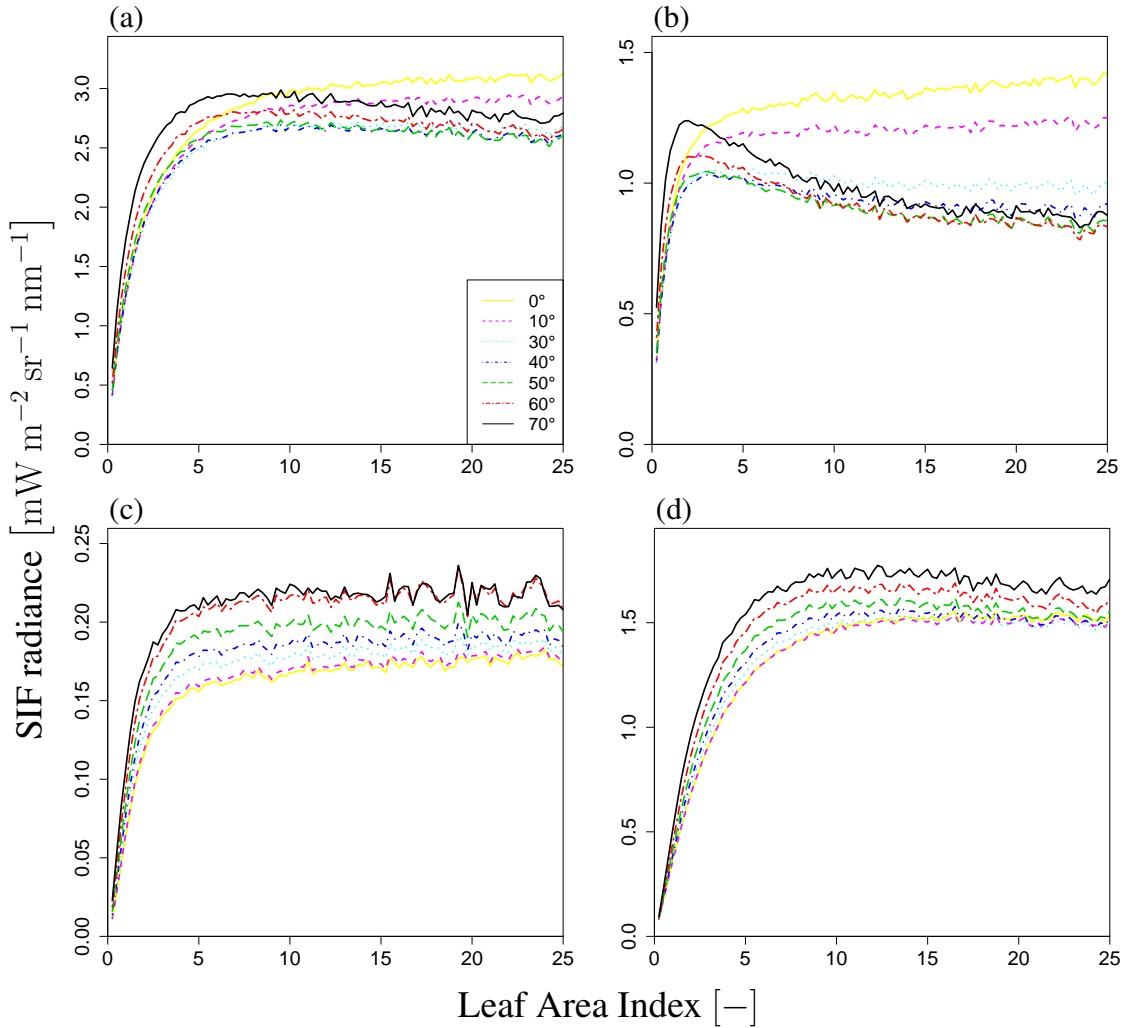

**Figure 15.** LAI dependence of SIF radiance. (a) Total radiance $I$, (b) direct radiance from sunlit leaves, (c) direct radiance from shaded leaves, and (d) radiance after multiple scatterings at SZA = 20° and SAA = 180° (backward direction). Each line represents a different VZA value (0° – 70°). SIF radiance increases with LAI and then becomes saturated in most cases. However, when VZA is large (e.g., black and red lines), only the direct radiance from sunlit leaves decreases with an increase in LAI, different from the forward direction case.

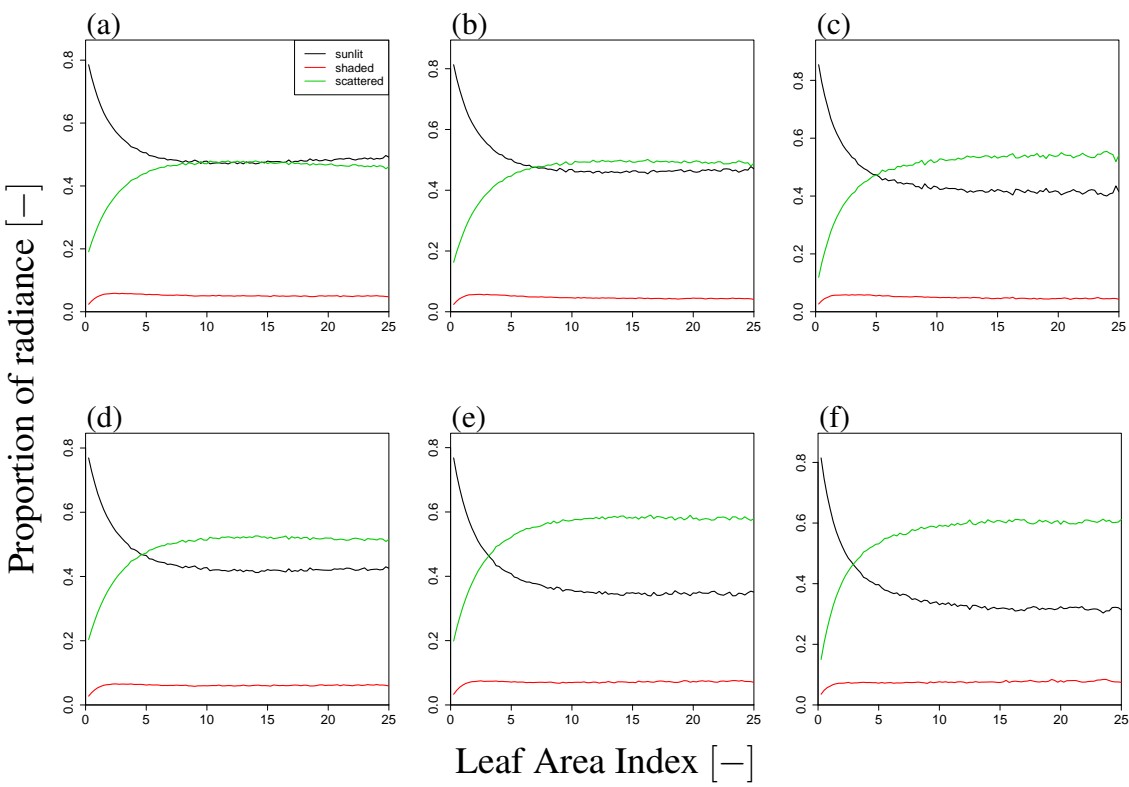

**Figure 16.** Proportion of SIF radiance in LAI variation. Upper figures (a) – (c) and lower figures (d) – (f) indicate results in forward and backward directions, respectively, for different VZA values (10° (left), 30° (center), and 50° (right)). The contribution of shaded leaves is small and the contribution of scattered radiance increases with LAI and VZA. Additionally, in the backward direction, the contribution of scattered radiation to SIF is larger than in the forward direction, similar to the angular dependency.

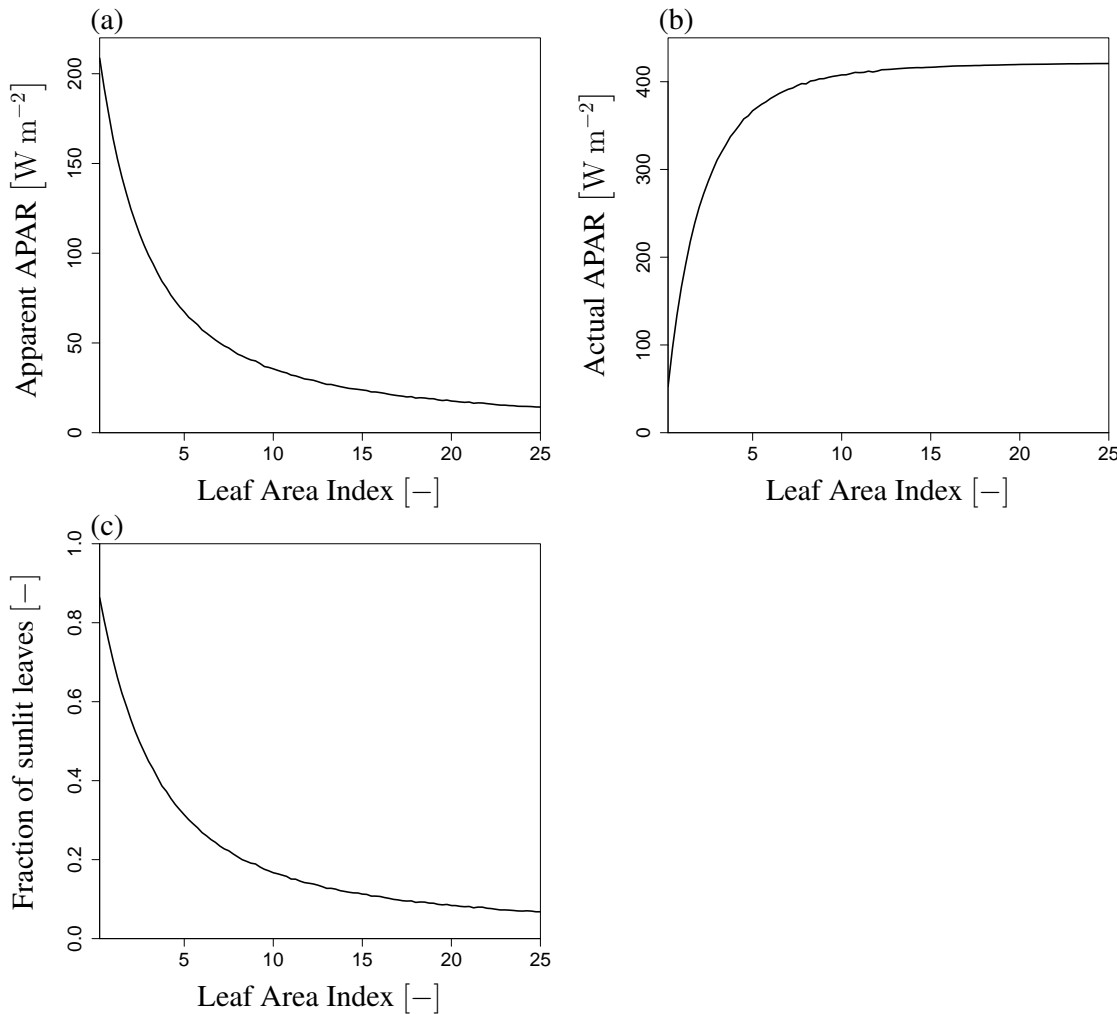

**Figure 17.** Variation of apparent and actual APAR and fraction of sunlit leaves with respect to LAI. (a) Apparent APAR, (b) Actual APAR, and (c) Fraction of sunlit leaves at SZA $= 20°$. These variables are not affected by the view direction. $\text{APAR}_{\text{app}}$ and $F_{\text{sun}}$ decrease exponentially and $\text{APAR}_c$ increases and then becomes saturated with an increase in LAI.

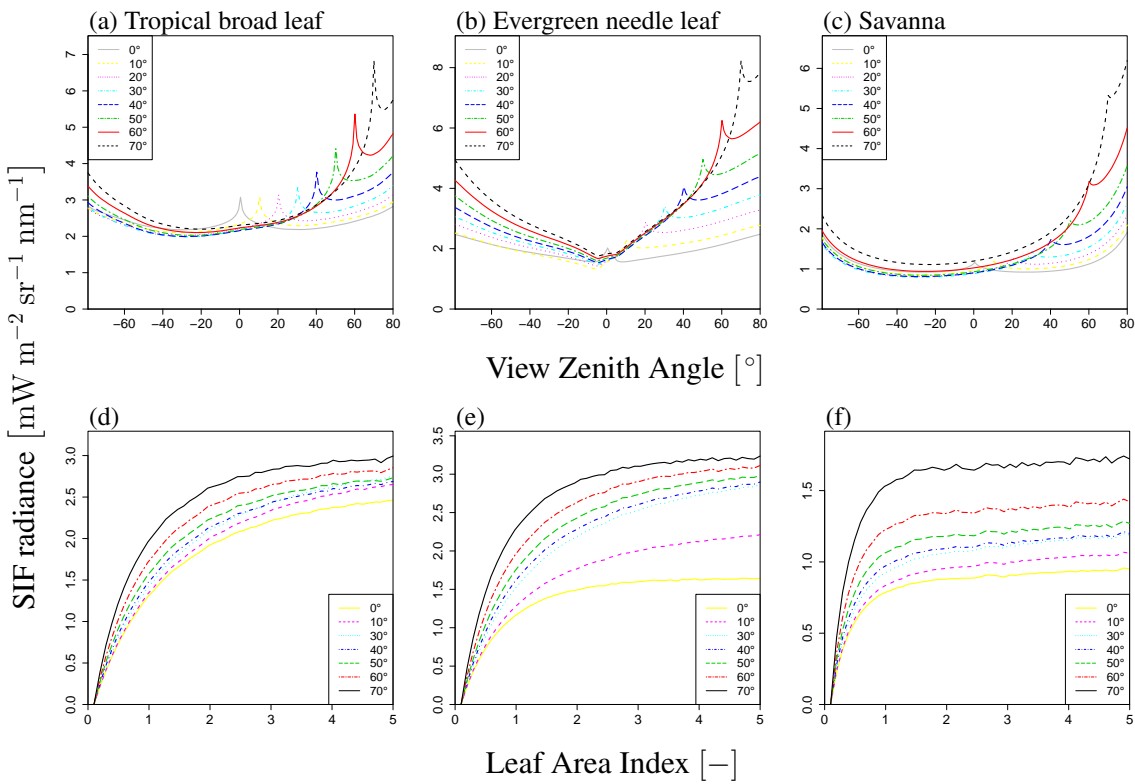

**Figure 18.** Comparison of SIF radiance with different landscape;. Upper and lower figures indicate VZA dependency (LAI = 3.0) and LAI dependency (VZA=20°), respectively. Each line represents a different SZA value (0° – 70°).

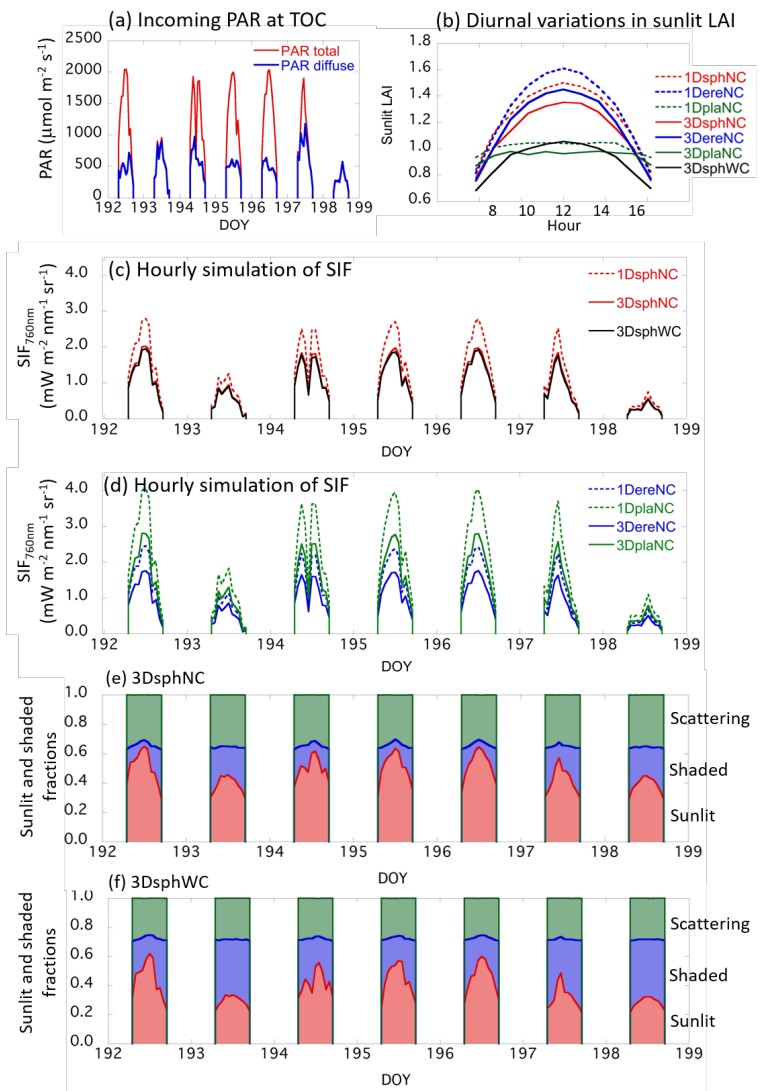

**Figure 19.** Summary of the diurnal SIF variations with an actual incident PAR data. (a) incoming total and diffuse PAR at the top of canopy (TOC). The observed PAR data in photon unit $\left(\mu\,mol\,m^{-2}\,s^{-1}\right)$ are converted to the unit of energy unit $\left(W\,m^{-2}\right)$ for the SIF simulation. (b) diurnal variations in sunlit LAI for different canopy structures. In all scenarios, total LAIs were set to 3. (c) simulated SIF radiances in hourly time scale for 1DsphNC, 3DsphNC, 3DsphWC. (d) simulated SIF radiances in hourly time scale for 1DereNC, 1DplaNC, 3DereNC, 3DplaNC. Dotted lines are 1D scenarios and solid lines are the results of 3D scenarios. (e) the fraction of sunlit, shaded and scattering contributions in the simulated total SIF radiance for 3DsphNC (no-clumping). (f) the fraction of sunlit, shaded and scattering contributions in the simulated total SIF radiance for 3DsphWC (within-crown clumping).

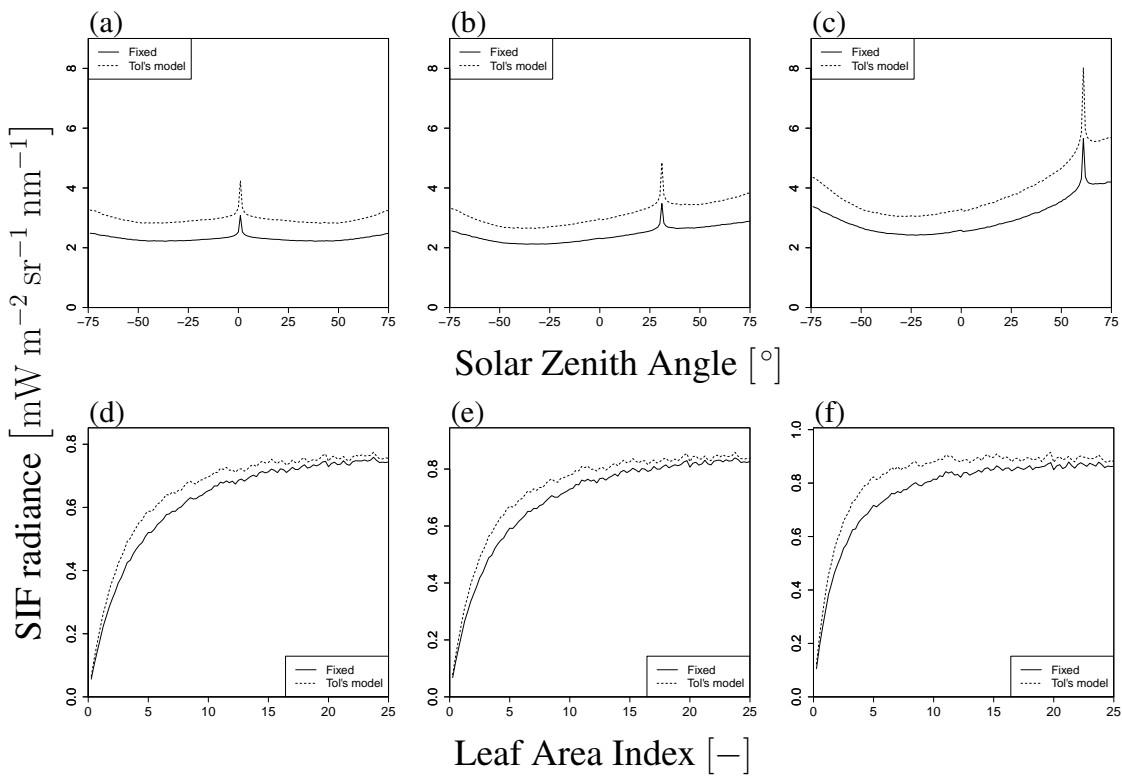

**Figure 20.** Comparison of SIF radiance with different $\phi_f$ (constant and from Tol's model). Upper and lower figures indicate SZA dependency (LAI = 3.0) and LAI dependency (SZA = 20°), respectively. The VZA values are 0° (**a** and **d**), 30° (**b** and **e**), and 60° (**c** and **f**). The solid line indicates the case of constant $\phi_f$ (= 0.01). The dashed line indicates the result of Tol's model, where $\phi_f$ depends on $\mathrm{APAR_L}$, as shown in Fig. 8.