# Peer review of "FLiES-SIF ver. 1.0: Three-dimensional radiative transfer model for estimating solar induced fluorescence"

_Geoscientific Model Development, 2020_

## Referee Comment (RC1) · Anonymous Referee #1 · 25 Mar 2020

General comments This manuscript by Sakai et al. presents a new 3D canopy radiative transfer model for chlorophyll fluorescence (SIF). Considering the rapid advances of remote sensing of SIF during the last several years, especially on the impacts of canopy structure on SIF, the 3D SIF model FLiES-SIF developed is therefore relevant and it will be of interest for the scientific community working on modeling of SIF and also for the user community. The manuscript is a nice addition to the current body of literature and I think it is worth publishing. However, a few comments may be taken into account to improve the manuscript.

My first concern is the comparison of the new FLiES-SIF 3D model to some field data,

at least to some other model simulations, such as DART. I knew some groups are doing bi-directional SIF measurements in the field. These data may be used to validate the new model to some extent. Also, the comparison to DART would also give some hints on the performances of FLiES-SIF.

Meanwhile, in the section of Introduction, it seems that there are missing in some new advances and recent publications on how canopy structure impacts the top-of-canopy SIF during the last two years. The authors may consider including them.

Several points:

1. L58: The expression of "At present, the Discrete...is the only available 3D model" is not clear. Is the DART model the only available 3D model to simulate SIF or other processes? As far as I know, there are other models that can simulate SIF, such as FluorFLIM (Zarco-Tejada et al, 2013), FluorFLIGHT (Hernández-Clemente et al., 2017). Are these not 3D models?

References: Hernández-Clemente R, North P R J, Hornero A and Zarco-Tejada P J. 2017. Assessing the effects of forest health on sun-induced chlorophyll fluorescence using the FluorFLIGHT 3-D radiative transfer model to account for forest structure. Remote Sensing of Environment 193: 165-179.[doi:10.1016/j.rse.2017.02.012]. Zarco-Tejada, P., Suárez, L., & Gonzalez-dugo, V. (2013). Spatial Resolution Effects on Chlorophyll Fluorescence Retrieval in a Heterogeneous Canopy Using Hyperspectral Imagery and Radiative Transfer Simulation. IEEE Geoscience and Remote Sensing Letters, 10, 937-941. doi:10.1109/LGRS.2013.2252877

2. It is not clear how to calculate APARC in Eq. (2)? Please add some information of the method.

3. According to the phase function for SIF emissions in Eq. (12), SIF emissions are calculated from the adaxial and abaxial sides of a leaf separately, which indicates hemisphere integration. But in Eq. (4), the normalization factor is $4\pi$. Should it be $4\pi$ or

$2\pi$?

4. Eq. (10) demonstrated the leaf-level SIF emission. Since you have already used a leaf level SIF model (FluoMODleaf) to derive the fraction of SIF emission from adaxial and abaxial side of leaves, I am curious why don't you use this model to simulate SIF emissions at leaf level?

5. To reduce time, the simulation of SIF direct emission (Eq. (5)) and APARL (Eq. (11)) both follow the Beer-law instead of using the backward ray tracing method. Regarding to the simulations, are there a large differences between the two methods? The assumption of a homogeneous layer should be made to apply the Beer-law attenuation. Does that indicate the model is not a real "3D" model in the conventional sense?

6. How do you calculate the scattering parameter $w_{i,j}$ in Eq. (14)?

7. Please add the description of the parameter GS in Eq. (16).

8. The authors have simulated the broadband SIF and considered the multi-scattering effect in the near-infrared spectral domain. Have you considered the re-absorption effect of SIF in the red spectral range?

9. Figure 3: The arrows in Fig 3. did not point the voxels clearly.

10. To exhibit the variation of SIF with wavelengths clearly, it would be good for the Fig 8. and Fig 9. to be transformed into three-dimensional images.

11. L178: replace "The SIF radiance emitted..." by "The scattered SIF radiance emitted...".

12. L315: replace "contribute" by "contributes".

---

## Referee Comment (RC2) · Anonymous Referee #2 · 6 Apr 2020

This manuscript presents the FLiES-SIF ver. 1.0 model, a 3-D radiative transfer model for SIF, and provides sensitivity analyses of the model to LAI, VZA, and fluorescence yield. Despite the recent advances in remote sensing of SIF, the impact of 3-D canopy structure on SIF signal is not well understood. Thus, the radiative transfer model for 3-D canopy and the sensitivity analysis presented here is of interest to the SIF community and has the potential to improve the estimation of SIF from satellite platforms.

Overall, the efforts to simulate SIF in 3-D canopy are important. However, the manuscript needs to be improved to make it a better contribution to the community. Below listed the major concerns, followed by some detailed comments.

1/ The lack of leaf physiology. This model does not have a leaf fluorescence module that is based on key parameters that control the SIF emission. For example, Vcmax or Chl. This is particularly important for the estimation of SIFyield, as the relationship between SIFyield and APAR depends on Vcmax and Chl. Without the leaf physiology component, the model has a limited use for the correction of satellite SIF data.

While running 3-D radiative transfer model can be computationally expensive, our computers have also advanced quite significantly in the past 10 years since the first publication of FLiES (Kobayashi and Iwabuchi 2008). Adding a photosynthesis module to the model will put this model to a higher level.

If adding a photosynthesis module is difficult at this point, I would at least ask the authors run a much more extensive simulation in SCOPE to estimate the potential uncertainties in Figure 7.

2/ The lack of details for the readers to evaluate and (potentially) reproduce. Information on the following processes / models / parameters are needed:

- How were the SCOPE runs conducted? What are the parameters used in the SCOPE run?
- In Section 3.4., the authors mentioned that they used the Farquhar model to obtain data on the photosynthetic rate. How exactly was it done?
- Section 2.2.2., how was fs determined? Where was the data source that gave the full SIF spectra?

3/ Overall structure of section 2. It would make the readers' job easier if there is an overarching paragraph and a diagram (not Figure 2) showing each component of the model, and how they are interconnected. For example, provide the description of canopy representation and some basic assumptions (e.g., turbid media) at the beginning of the section, as this information is essential for readers to understand some of the equations. This section in the current form reads like that each subsection is disconnected.

4/ The unit for SIF. Whenever SIF from a specific wavelength is simulated, it should be in the unit for spectral radiance, which is mw/m2/sr/nm. Check figures like Figure 9.

5/ The benefit of 3-D modeling. Just for the benefit for the readers, can you provide some sensitivity analysis of SIF simulations to different canopy structures? This is perhaps one of the key novelties compared with 1-D models.

General comments:

1/ Please provide continuous line numbers, instead of numbers every five lines.

2/ L3: have revealed instead of have been revealed

3/ L9-10: "due to the lack of complexity" should describe 1-D models, not the 3-D models.

4/ L11: the → a.

5/ Line 33: Frankenberg et al. 2011 used GOSAT, not GOSAT 1&2.

6/ Line 49: "fluorescence signals enhanced by …". Be more specific, is it total fluorescence signal, or the fluorescence signal observed by the sensor? The former is weakened by reabsorption during multiple scattering.

7/ L52: the causality – this word here is confusing.

8/ L58: DART-SIF is not the only available 3D model. There is at least also FluorFlight and FluorWPS.

9/ L83: top of canopy instead of atmosphere?

10/ L96: this sentence needs rewording. If the canopy is sparse, we would expect less attenuation. Do you mean more attenuation by the trunk?

11/ L99: forcing leaves to absorb all the photons does not make much biological sense here. Even for tropical forest, fPAR is 0.99 not 1. Please clarify.

12/ L123: what does it mean by "negligibly small"? Please quantify.

13/ L128: $d\Omega_L$ is redundant as you have $d\theta_L d\varphi_L$

14/ L141: should be $G(\Omega_L)+G(\Omega_j)$?

15/ L156: The integration should probably be for $\exp(-\tau s(\theta,\varphi))|\Omega \cdot \Omega_L|\sin\theta d\theta d\varphi$, please check this equation.

16/ L185: Write down this equation as some readers may not have access to the original paper.

17/ L251: Why is this limited to sunlit condition? Seems a spherical integration is also needed for shaded condition.

18/ L253: Is there a test on how well this method performs? It seems to me large zenith angles are underrepresented. How about a simulation test: Do a more precise numerical integration (e.g., average of 50 directions) and compare the result with the result from their proposed method (average of the five selected angles).

19/ L282: the fluorescence quantum efficiency of 0.04 seems to be too high. SCOPE used to have it as 0.02 and has to change it to 0.01 because the simulated SIF values were too high when using 0.02.

20/ L301: "shows the" instead of "shows that the"

21/ L338: $APAR_{app}$ instead of $APAR_c$?

22/ L363: the index for the equation is missing

23/ L372: in instead of inn

24/ L430: "the proposed model can …." It has the potential but it cannot do what is stated in this sentence as of now because the lack of leaf physiology.

25/ Figure 17: The sequence of upper and lower panels in the caption is not consistent with the figure: "Upper and lower figures indicate SZA dependency (LAI = 3.0) and LAI dependency (SZA=20°), respectively

---

## Author Response (AR1)

Response to the comments of two reviewers.

Dear editors,

Thank you very much for your comments on our manuscript and your sincere efforts in constructing a decision report. The comments and suggestions made by the two reviewers have been very useful in improving our manuscript. We have revised the manuscript following careful consideration of the reviewers' comments. In the revised manuscript, rewritten and additional sentences are indicated in red and blue, respectively. We hope the revised manuscript is now suitable for publication in Geoscientific Model Development. We look forward to your favorable consideration. Our responses to the reviewers' individual comments and questions are given below.

Reviewer #1

[Comment] My first concern is the comparison of the new FLiES-SIF 3D model to some field data, C1 at least to some other model simulations, such as DART. I knew some groups are doing bi- directional SIF measurements in the field. These data may be used to validate the new model to some extent. Also, the comparison to DART would also give some hints on the performances of FLiES-SIF.

[Response] Thank you for pointing out. We have added the new section to require your suggestion (see Sect. 3.2 on page 14).

[Comment] Meanwhile, in the section of Introduction, it seems that there are missing in some new advances and recent publications on how canopy structure impacts the top-of-canopy SIF during the last two years. The authors may consider including them.

[Response] Thank you for pointing out. We have added the new reference information (Line 52 on page 2).

Several points:

[Comments] 1. L58: The expression of "At present, the Discrete. . .is the only available 3D model" is not clear. Is the DART model the only available 3D model to simulate SIF or other processes? As far as I know, there are other models that can simulate SIF, such as FluorFLIM (Zarco-Tejada et al, 2013), FluorFLIGHT (Hernández-Clemente et al., 2017). Are these not 3D models?

References: Hernández-Clemente R, North P R J, Hornero A and Zarco-Tejada P J. 2017. Assessing the effects of forest health on sun-induced chlorophyll fluorescence using the FluorFLIGHT 3-D radiative transfer model to account for forest structure. Remote Sensing of Environment 193: 165-179.[doi:10.1016/j.rse.2017.02.012]. Zarco-Tejada, P., Suárez, L., & Gonzalez-dugo, V. (2013). Spatial Resolution Effects on Chlorophyll Fluorescence Retrieval in a Heterogeneous Canopy Using Hyperspectral Imagery and Radiative Transfer Simulation. IEEE Geoscience and Remote Sensing Letters, 10, 937-941. doi:10.1109/LGRS.2013.2252877

**[Response]** Thank you for pointing us in the direction of these studies. We have read these articles and added the reference information on Line 63 (page 3).

**[Comments]** 2. It is not clear how to calculate APARC in Eq. (2)? Please add some information of the method.

**[Response]** In the previous manuscript, the method of computing APARc was not described in detail (Lines 100–101 in the previous manuscript). APARc is independently computed by the FLiES-SIF APAR computation module, which is basically the same as the numerical scheme used in FLiES version 2.4 (Kobayashi, Hideki. (2019, August 6), Zenodo. http://doi.org/10.5281/zenodo.3586814). In the revised manuscript, we have added a detailed description of the model framework (e.g., Fig. 1 and Sect. 2.1 in the revised manuscript) and explain how to compute APARc in Sect. 2.1.3 (Lines 130–146). A flowchart of the simulation process is given in the new Fig. 1(b). In summary, APARc is computed by the radiative transfer computation in the broad PAR domain (400–700 nm) before the spectral SIF radiance is simulated. This APARc is used to re-scale the SIF radiance under the actual APAR conditions.

**[Comments]** 3. According to the phase function for SIF emissions in Eq. (12), SIF emissions are calculated from the adaxial and abaxial sides of a leaf separately, which indicates hemi-sphere integration. But in Eq. (4), the normalization factor is $4\pi$. Should it be $4\pi$ or $2\pi$?

**[Response]** Thank you for pointing out this mistake. The correct normalization factor in Eq. (5) (page 8 in revised manuscript) is $2\pi$. We have modified this equation. We checked all other equations carefully and found that they were correct. In addition, the source code was correctly described. Thus, this error would not have affected the simulation results.

**[Comments]** 4. Eq. (10) demonstrated the leaf-level SIF emission. Since you have already used a leaf level SIF model (FluoMODleaf) to derive the fraction of SIF emission from adaxial and abaxial side of leaves, I am curious why don't you use this model to simulate SIF emissions at leaf level?

**[Response]** In our modeling, we use the FluorMODleaf model to derive the fraction of SIF emissions from the adaxial and abaxial sides of leaves (in the emission phase function) and the spectral composition of SIF, i.e., the factor $f_s$ in Eq. (11) (Eq. (10) in the previous manuscript). The broadband fluorescence energy is determined from APAR and the SIF yield computed by the model of Tol et al. (2014) and Farquhar's model (Farquhar et al., 1980). This enables us to couple the leaf traits (Vcmax, Jmax) and leaf physiological responses to environmental conditions (temperature, humidity, $pCO_2$). Our approach is similar to that of the SCOPE model (Tol et al., 2009). In the revised manuscript, we have added a description of how we use FluorMODleaf and how we combine the leaf physiology module with the radiative transfer module (see Sect. 2.1.5 on page 6).

*"To simulate the spectral SIF, the spectral composition of SIF must be known. Our approach is similar to that used in the SCOPE model (Tol et al., 2009). We derived the spectral composition from the FluorMODleaf model (Zarco-Tejada et al., 2006; Pedrós et al., 2010). The calculated leaf-level spectral SIF radiance variations given by FluorMODleaf were normalized to determine the fraction of SIF at wavelength , λ, fs (mW m−2 sr−1), with respect to the broadband (W m−2). That is, we only used the fraction of spectral composition from the FluorMODleaf model. The radiance was then determined from APAR and $\phi_f$, which varies with environmental conditions and leaf traits such as the maximum carboxylation capacity, Vcmax, used in the photosynthesis model."* (Lines 182-188 on page 6)

**[Comments]** 5. To reduce time, the simulation of SIF direct emission (Eq. (5)) and APARL (Eq. (11)) both follow the Beer-law instead of using the backward ray tracing method. Regarding to the simulations, are there a large differences between the two methods? The assumption of a homogeneous layer should be made to apply the Beer-law attenuation. Does that indicate the model is not a real "3D" model in the conventional sense?

**[Response]** The attenuation function in Eqs. (6) and (12) (Eqs. (5) and (11) in the previous manuscript) are the same. FLiES-SIF is a 3D model in which individual trees are explicitly defined in a certain landscape. To clarify the 3D feature of the attenuation function, we modified Eq. (6) (Eq. (5) in the previous manuscript) as follows:

$$\exp(-\tau_\sigma) = exp\left(-\sum_i u_i \gamma_i G_{\sigma,i} s_i\right)$$

where the optical thickness $\tau_\sigma$ is computed as the sum of light paths of the *i*th tree from the emission point to the view direction. The same analogy can be applied to Eq. (12) (Eq. (11) in the previous manuscript).

We have modified the descriptions of these equations as follows:

"where ui, si, Gσ,i, and γi are the leaf area density, path length, mean leaf projection area, and clumping index of the ith tree. They are aggregated over the trees located in the light path between the emission point to the top of canopy in the view direction, respectively." (Lines 234–236 on page 8)

**[Comments]** 6. How do you calculate the scattering parameter wi,j in Eq. (14)?

**[Response]** As described in the manuscript, $w_{i,j}$ is the weight of a photon with an initial weight of $w_0$ (Eq. 11). $w_{i,j}$ is computed by multiplying the photon weight of the previous scattering order by $w_{SIF}$ ($=r_{SIF} + t_{SIF}$).

**[Comments]**7. Please add the description of the parameter GS in Eq. (16).

**[Response]** We have added a description of GS. This is the mean leaf projection area, as defined in Eq. (7). (Line 320 on page 12)

**[Comments]**8. The authors have simulated the broadband SIF and considered the multi-scattering effect in the near-infrared spectral domain. Have you considered the re-absorption effect of SIF in the red spectral range?

**[Response]** The current FLiES-SIF model takes the re-absorption effect of the emitted SIF into account. In the photon tracing, when the emitted fluorescence light hits other leaves, it is absorbed or scattered. In the red spectral domain, because the chlorophyll absorption is high, the leaf reflectance and transmittance are lower than in the near-infrared domain and more fluorescence light is absorbed. This process is considered in the variable weighting of the photons, $w_{i,j}$, in Eq. (15). In the revised manuscript, we have added a new Sect. 2.1 to introduce the overall framework. The scattering and re-absorption processes in the FLiES-SIF model are now described as follows:

*"The scattering and re-absorption of emitted fluorescence light must also be considered to identify the relationship between the fluorescence emitted by the chloroplasts and the top-of-canopy outgoing fluorescence 100 radiance (Porcar-Castell et al., 2014). Several recent studies have worked on the quantification of the impact and modeling of scattering and absorption effects from the leaf scale (e.g., Agati et al. (1993); van der Tol et al. (2019)) to the canopy scale (e.g., Romero et al. (2018)). Multiple scatterings and re-absorption among leaves, trunks, and soil background can be numerically simulated using unbiased and efficient approaches (Kobayashi and Iwabuchi, 2008)."* (Lines 98–103 on page 4)

**[Comments]**9. Figure 3: The arrows in Fig 3. did not point the voxels clearly.

**[Response]** Thank you for pointing out this issue. We have corrected Fig. 3 (Page 28).

**[Comments]**10. To exhibit the variation of SIF with wavelengths clearly, it would be good for the Fig 8. and Fig 9. to be transformed into three-dimensional images.

**[Response]** Thank you for this suggestion. These figures have been transformed to 3D images (Fig. 9 on page 33).

**[Comments]**11. L178: replace "The SIF radiance emitted..." by "The scattered SIF radiance emitted. . .".

**[Response]** Thank you for this suggestion. This has been modified accordingly. (Line 289 on page 10)

**[Comments]**12. L315: replace "contribute" by "contributes".

**[Response]** Thank you for pointing out this error. This has been corrected. (Line 431 on page 15)

Reviewer #2

[Comments] This manuscript presents the FLiES-SIF ver. 1.0 model, a 3-D radiative transfer model for SIF, and provides sensitivity analyses of the model to LAI, VZA, and fluorescence yield. Despite the recent advances in remote sensing of SIF, the impact of 3-D canopy structure on SIF signal is not well understood. Thus, the radiative transfer model for 3-D canopy and the sensitivity analysis presented here is of interest to the SIF community and has the potential to improve the estimation of SIF from satellite platforms.

Overall, the efforts to simulate SIF in 3-D canopy are important. However, the manuscript needs to be improved to make it a better contribution to the community. Below listed the major concerns, followed by some detailed comments.

**[Comments]**1/ The lack of leaf physiology. This model does not have a leaf fluorescence module that is based on key parameters that control the SIF emission. For example, Vcmax or Chl. This is particularly important for the estimation of SIFyield, as the relationship between SIFyield and APAR depends on Vcmax and Chl. Without the leaf physiology component, the model has a limited use for the correction of satellite SIF data.

While running 3-D radiative transfer model can be computationally expensive, our computers have also advanced quite significantly in the past 10 years since the first publication of FLiES (Kobayashi and Iwabuchi 2008). Adding a photosynthesis module to the model will put this model to a higher level.

If adding a photosynthesis module is difficult at this point, I would at least ask the authors run a much more extensive simulation in SCOPE to estimate the potential uncertainties in Figure 7.

**[Response]** In accordance with comment 3 by this reviewer, we have substantially improved Sect. 2. The FLiES-SIF model does indeed have a leaf physiology module, although our leaf-scale module itself is not a new model and is based on two existing models (those of Tol et al. (2014) and Farquhar et al. (1980)). This is why we did not include a detailed description of leaf physiology and fluorescence in the original paper. In the revised manuscript, we have added a description of how we incorporated the leaf physiology models in Sects. 2.1.3 (Simulation flow) and 2.1.5 (Computation of leaf level fluorescence yield). As described in these subsections, we

created a look-up table of the SIF yield $\varphi_f$ under various environmental conditions and leaf traits (such as maximum carboxylation capacity, Vcmax). In the FLiES framework, there is a module that computes the interrelations among the energy balance (leaf temperature), stomata, and photosynthesis based on the CANOAK model (Bakdocchi and Harkey, 1995). However, this would entail a greater computational load and require further input variables. Thus, in the current FLiES-SIF, we used the following assumptions to obtain reasonable photosynthesis simulation results. First, the leaf temperature was assumed to be the same as the surface air temperature. This is usually acceptable, except in very dry conditions where the stomata are almost closed in daytime. The other assumption concerns the stomata modeling. The FLiES-SIF module does not explicitly use the stomata model. Rather, the consequences of the stomata activity, i.e., down-regulation of intercellular partial $CO_2$ pressure (ipCO$_2$), were modeled as a function of the vapor pressure deficit (VPD). (Sect. 2.1.3 on page 5 and Sect. 2.1.5 on page 6)

**[Comments]**2/ The lack of details for the readers to evaluate and (potentially) reproduce. Information on the following processes / models / parameters are needed:

- How were the SCOPE runs conducted? What are the parameters used in the SCOPE run? ⌷

**[Response]** Thank you for your comments. We added the information of leaf physiology module of FLiES-SIF in improved model description section (especially, Section 2.1)

- In Section 3.4., the authors mentioned that they used the Farquhar model to obtain data on the ⌷photosynthetic rate. How exactly was it done? ⌷

**[Response]** Thank you for your comments. I apologize for a confusion. We used the Farquhar model to obtain the tentative photosynthetic yield to derive the phi_f. This has been corrected in the revised manuscript. (Lines 564-566 on page 19).

- Section 2.2.2., how was fs determined? Where was the data source that gave the full SIF spectra? ⌷

**[Response]** The leaf module of FluorMOD was used to determine fs and the leaf-level SIF spectra. We derived the leaf SIF spectra information from the FluorMODleaf model (Zarco-Tejada et al., 2006; Pedrós et al., 2010). The calculated leaf-level spectral SIF radiance variations were then normalized to determine the fraction of SIF at wavelength $\lambda$, fs (mW m$^{-2}$ sr$^{-1}$), with respect to the broadband SIF (W m$^{-2}$). That is, we only used the fraction of the spectral composition from the FluorMODleaf model. The radiance was then determined from APAR and $\varphi_f$, which varies with environmental conditions and leaf traits such as the maximum carboxylation capacity, Vcmax, used in the photosynthesis model. We have added the above description in Sect. 2.1.5. (Lines 182–188 on page 6)

**[Comments]** 3/ Overall structure of section 2. It would make the readers' job easier if there is an

overarching paragraph and a diagram (not Figure 2) showing each component of the model, and how they are interconnected. For example, provide the description of canopy representation and some basic assumptions (e.g., turbid media) at the beginning of the section, as this information is essential for readers to understand some of the equations. This section in the current form reads like that each subsection is disconnected. [SEP]

[Response] Section 2 has been substantially revised and improved. As suggested, we have added a new subsection 2.1 (General outline of FLiES-SIF), in which we summarize the overall framework of the FLiES modeling. Newly added Fig. 1(a) shows each radiative transfer component and how they are related. In the previous manuscript, the canopy representation was described in Sect. 2.4. This description has been moved to subsection 2.1.2 (Canopy structure represented by FLiES-SIF). The basic assumptions made for the crown volumes (e.g., turbid media, clumping, and leaf area density distributions) are also described in this section. We have also added a flowchart illustrating the simulation process and the major input variables used in the model (Fig. 1(b)). This describes how some basic information (forest structures and leaf physiology) is derived from the input variables and how the FLiES-SIF model proceeds. (Pages 3–6)

[Comments]4/ The unit for SIF. Whenever SIF from a specific wavelength is simulated, it should be in the unit for spectral radiance, which is mw/m2/sr/nm. Check figures like Figure 9. [SEP]

[Response] Thank you for this comment. This has been corrected in the revised manuscript. (Figs. 11, 14, 15, and 20)

[Comments]5/ The benefit of 3-D modeling. Just for the benefit for the readers, can you provide some sensitivity analysis of SIF simulations to different canopy structures? This is perhaps one of the key novelties compared with 1-D models. [SEP]

[Response] Thank you for this suggestion. We have added simulations to different canopy structures and a comparison of 1D and 3D modeling in subsections 3.4 and 3.5, and the figures therein. (Pages 18-19)

General comments: [SEP]

[Comments]1/ Please provide continuous line numbers, instead of numbers every five lines. [SEP]

[Response] Thank you for your comment. However, we have compiled the manuscript using the LaTeX package supplied by Copernicus Publications. Thus, we cannot change the line number format.

**[Comments]**2/ L3: have revealed instead of have been revealed⌷SEP⌷

**[Response]**Thank you for pointing out this mistake. This has been corrected. (Line 3 on page 1)

**[Comments]**3/ L9-10: "due to the lack of complexity" should describe 1-D models, not the 3-D models. ⌷SEP⌷

**[Response]**Thank you for pointing out this mistake. This has been corrected. (Lines 10–11 on page 1)

**[Comments]**4/ L11: the→a.⌷SEP⌷

**[Response]** Thank you for pointing out this mistake. This has been corrected. (Line 12 on page 1)

**[Comments]**5/ Line 33: Frankenberg et al. 2011 used GOSAT, not GOSAT 1&2.

**[Response]** Thank you for pointing out this mistake. This has been corrected. (Line 37 on page 2)

**[Comments]**6/ Line 49: "fluorescence signals enhanced by ...". Be more specific, is it total fluorescence signal, or the fluorescence signal observed by the sensor? The former is weakened by reabsorption during multiple scattering.

**[Response]** Thank you for pointing out this omission. This refers to the fluorescence signals observed by sensors, as now specified in the manuscript. (Line 53 on page 2)

**[Comments]**7/ L52: the causality – this word here is confusing.

**[Response]** Thank you for pointing out this error. The phrase "causality of directional canopy SIF" was inappropriate. We have changed this to read "mechanism of anisotropic light interactions such as scattering and absorption in plant canopies."(Line 57 on page 2)

**[Comments]**8/ L58: DART-SIF is not the only available 3D model. There is at least also

FluorFlight and FluorWPS.

**[Response]** Thank you for pointing out this error. We have investigated these models and have added the appropriate references. (Line 63 on page 3)

**[Comments]** 9/ L83: top of canopy instead of atmosphere?

**[Response]**. Thank you for pointing out this mistake. This has been corrected. (Line 190 on page 7)

**[Comments]** 10/ L96: this sentence needs rewording. If the canopy is sparse, we would expect less attenuation. Do you mean more attenuation by the trunk?

**[Response]** Thank you for this comment. We meant to refer to "transmitted PAR" rather than "attenuated." We have replaced "attenuated" with "transmitted." (Line 204 on page 7)

**[Comments]** 11/ L99: forcing leaves to absorb all the photons does not make much biological sense here. Even for tropical forest, fPAR is 0.99 not 1. Please clarify.

**[Response]**. This is a variance reduction technique for the Monte Carlo ray tracing proposed in this study. The proposed method, as noted by the reviewer, artificially enhances $f_{apar}$ in the Monte Carlo simulation. However, the simulated SIF radiance is later scaled by the actual PAR (APARc) (please see the new Fig. 1(b) in the revised manuscript). By applying this scaling, the simulated SIF will not be biased, even if we force all photons to hit leaves.

**[Comments]** 12/ L123: what does it mean by "negligibly small"? Please quantify.

**[Response]** Thank you for your comment. The phrase "negligibly small" was inappropriate. We have changed this to read "when the hotspot effect is not considered." (Lines 231–232 on page 8)

**[Comments]** 13/ L128: $d\Omega_L$ is redundant as you have $d\theta_L d\varphi_L$

**[Response]** Thank you for this comment. To retain consistency in the parameter definitions, we have removed $d\theta_L d\varphi_L$. (Equation (9) on page 8)

**[Comments]** 14/ L141: should be $G(\Omega_L)+G(\Omega_j)$?

**[Response]** Thank you for pointing out this mistake. This has been corrected. (Equation (10) on page 9)

**[Comments]** 15/ L156: The integration should probably be for $\exp(-\tau s(\theta,\varphi))|\Omega \cdot \Omega_L|\sin\theta d\theta d\varphi$, please check this equation.

**[Response]** Thank you for this comment. Indeed, you are correct. We have modified the equation accordingly. (Equation (12) on page 9)

**[Comments]** 16/ L185: Write down this equation as some readers may not have access to the original paper.

**[Response]** The description is not straightforward. This equation is the same as Eq. (16) in Kobayashi and Iwabuhi (2008). We have rephrased this sentence as follows: "E quation (15) is exactly the same as the multiple scatterins in the shortwave radiative transfer (Kobayashi and Iwabuchi, 2008)." (Lines 295-296)

**[Comments]** 17/ L251: Why is this limited to sunlit condition? Seems a spherical integration is also needed for shaded condition.

**[Response]** This is correct. The spherical integration is necessary for both sunlit and shaded leaves. In the revised manuscript, we have removed the words "For the sunlit leaf condition." (Line 347 on page 12)

**[Comments]** 18/ L253: Is there a test on how well this method performs? It seems to me large zenith angles are underrepresented. How about a simulation test: Do a more precise numerical integration (e.g., average of 50 directions) and compare the result with the result from their proposed method (average of the five selected angles).

**[Response]** We have added the results of an accuracy assessment of this 5-angle approximation by comparing the reliable 10-degree samplings (9 zenith angles × 36 azimuth angles = 324 angle samplings). When the attenuation functions were computed by these two angle sampling approaches at 10000 randomly selected positions in the forest landscapes used in the sensitivity analysis described in Sect. 3, the mean absolute error of this approximation was 14.6% ($N = 10000$). We have added a description of the accuracy of this 5-angle assumption in Sect. 2.5 C.

"We tested the performance of this 5-angle assumption by comparing with 10-degree interval samplings (9 zenith and 36 azimuth angles = 324 angle sampilngs). When the attenuation functions were computed by these two angle samplings at 104 randomly selected positions in the forest landscapes used in the sensitivity analysis in section 3, the mean absolute error of this approximation was 14.6 % (N = 10000)." (Lines 340–343 on page 12)

Note that there was an error in the angle information in the previous manuscript ((0°, 0°), (45°, 0°), (45°, 90°), (45°, 180°), and (45°, 270°).). The correct zenith angles are (0°, 0°), (60°, 0°), (60°, 90°), (60°, 180°), and (60°, 270°). In the revised manuscript, this error has been corrected. (Line 339-340)

**[Comments]** 19/ L282: the fluorescence quantum efficiency of 0.04 seems to be too high. SCOPE used to have it as 0.02 and has to change it to 0.01 because the simulated SIF values were too high when using 0.02.

**[Response]** Thank you for this suggestion. As you stated, the fluorescence quantum efficiency of 0.04 was too high and not suitable. We have rerun FluorMODleaf with F = 0.01 and saved all updated values. Usually, the F value is linearly related to the leaf SIF radiance if all other parameters remain unchanged. As noted in Sect. 2.1.5 of the revised manuscript, we only used the leaf-level spectral SIF radiance to determine the fraction of spectral contributions. Thus, this change does not influence the subsequent sensitivity studies in Sect. 3. (Page 13)

**[Comments]**20/ L301: "shows the" instead of "shows that the"

**[Response]** Thank you for this comment. This has been corrected. (Line 417 on page 15)

**[Comments]**21/ L338: APARapp instead of APARc? SEP

**[Response]** Thank you for pointing this out. This has been corrected. (Line 454 on page 16)

**[Comments]**22/ L363: the index for the equation is missing

**[Response]** Thank you for pointing this out. This has been corrected. (Line 480 on page 17)

**[Comments]**23/ L372: in instead of inn

**[Response]** Thank you for pointing out this mistake. This has been corrected. (Line 489 on page 17)

**[Comments]**24/ L430: "the proposed model can ...." It has the potential but it cannot do what is stated in this sentence as of now because the lack of leaf physiology.

**[Response]** Thank you for this comment. As stated in our improved Sect. 2, the proposed model does in fact include a leaf physiology module.

**[Comments]**25/ Figure 17: The sequence of upper and lower panels in the caption is not consistent with the figure: "
[revised manuscript text omitted]

**2.3.1 Attenuation function**

The attenuation of SIF in the view direction  $\mathbf{\Omega}_\sigma$ is calculated by the attenuation function $\exp(-\tau_v)$. When the hotspot effect is  not considered, the attenuation function is expressed using the plant canopy gap fraction theory:

$$\exp(-\tau_\sigma) = \exp\left(-\sum_i u_i \gamma_i G_\sigma \int uds_{\sigma,i} s_i\right) \tag{6}$$

where  $u_i$, $s_i$, $G_{\sigma,i}$ and $\gamma_i$ are the leaf area density,  path length, mean leaf projection area, and clumping index of the $i$-th tree. They are aggregated over the trees located in the light path between the emission point to the top of canopy in the view direction, respectively. The path length, $s$, is a sum of canopy paths that penetrates through crown objects. The mean leaf projection area $G$ is a function of the leaf inclination angle distribution function $g_L$ and an arbitrary direction $\mathbf{\Omega}_\sigma$ (such as the sun direction $\mathbf{\Omega}_s$ or view direction $\mathbf{\Omega}_v$):

$$G_\sigma := G(\mathbf{\Omega}_\sigma) = \frac{1}{2\pi} \int_0^{2\pi} \int_0^{\frac{\pi}{2}} g_L(\mathbf{\Omega}_L) |\mathbf{\Omega}_L \cdot \mathbf{\Omega}_\sigma| d\mathbf{\Omega}_L d\theta_L d\phi_
[revised manuscript text omitted]
_{sun} &= \frac{1}{G_s} \lim_{\Delta L \to 0} \frac{\exp\left(-G_s \gamma L_p\right) - \exp\left(-G_s \gamma \left(L_p + \Delta L_p\right)\right)}{\Delta L} \\
&= \gamma \exp\left(-G_s \gamma L_p\right)
\end{aligned} \tag{17}
$$

355 where $L_p$ is the cumulative LAI at $v_0$ along the path of the sunlight and $G_S$ is a mean leaf projection area defined in Eq. 7. The leaf illumination status (sunlit or shaded) is then determined by a random number $R$:

$$
\begin{cases}
R \leq P_{sun} & \to \text{Sunlit leaf} \\
R > P_{sun} & \to \text{Shade leaf}
\end{cases} \tag{18}
$$

The leaf surface normal vector $\Omega_L$ is also required because the leaf-level SIF emission is related to APAR at the leaf surface ($APAR_L$). $APAR_L$ is computed from the cosine of the sunlight and leaf normal angles. Assuming the leaves are randomly 360 distributed, the azimuthal angle of the leaf surface normal $\phi_L$ can be determined by:

$$
\phi_L = 2\pi R \tag{19}
$$

For a given leaf angle distribution function $g_L := g\left(\theta_L\right)$, the zenith angle of the leaf surface normal $\theta_L$ can be determined by the rejection method. In the first step, $\theta_L$ is calculated using a random number:

$$
\theta_L = \frac{\pi}{2} R \tag{20}
$$

365 Then, $\theta_L$ is further evaluated using $g_L$:

$$
\begin{cases}
R \leq g_L \sin \theta_L & \to \text{select} \\

[revised manuscript text omitted]